# A high affinity switch for cAMP in the HCN pacemaker channels

Alessandro Porro [1,7], Andrea Saponaro [2,7], Roberta Castelli [1], Bianca Introini [1], Anahita Hafez Alkotob[1], Golnaz Ranjbari [1], Uta Enke[3], Jana Kusch [3], Klaus Benndorf [3], Bina Santoro[4], Dario DiFrancesco [1], Gerhard Thiel[5] & Anna Moroni [1,6] ✉

Binding of cAMP to Hyperpolarization activated cyclic nucleotide gated (HCN) channels facilitates pore opening. It is unclear why the isolated cyclic nucleotide binding domain (CNBD) displays in vitro lower affinity for cAMP than the full-length channel in patch experiments. Here we show that HCN are endowed with an affinity switch for cAMP. Alpha helices D and E, downstream of the cyclic nucleotide binding domain (CNBD), bind to and stabilize the holo CNBD in a high affinity state. These helices increase by 30-fold cAMP efficacy and affinity measured in patch clamp and ITC, respectively. We further show that helices D and E regulate affinity by interacting with helix C of the CNBD, similarly to the regulatory protein TRIP8b. Our results uncover an intramolecular mechanism whereby changes in binding affinity, rather than changes in cAMP concentration, can modulate HCN channels, adding another layer to the complex regulation of their activity.

Hyperpolarization and Cyclic Nucleotide (HCN)-gated channels are the molecular correlates of the "funny" pacemaker current ($I_f/I_h$), that contributes to the generation of the cardiac ($I_f$) and neuronal impulses ($I_h$)[1]. The regulation of HCN channels by the cyclic nucleotide plays important physiological and pathological roles in humans ranging from the control of heart rate in the sinoatrial node myocytes to the initiation of neuropathic pain in sensory neurons[2,3]. For this regulation HCN channels contain in their cytosolic C terminus, the cyclic nucleotide binding domain (CNBD), which binds to the second messenger cAMP.

The CNBD of HCN channels has been extensively studied since its first X-ray structure became available in 2003[4]. Eight beta strands and a short alpha helix (P) form the cAMP binding pocket, while alpha helices (αA-C) play two other important functions upon cAMP binding: αB and C move towards the cAMP binding pocket with αC acting as a "lid" that stabilizes ligand binding; αA moves upward and transmits the conformational movements in the binding domain via the C -linker to the transmembrane voltage sensor domain (VSD)[5–7]. As a result, cAMP

binding to the CNBD facilitates pore opening by shifting the voltage-dependency of HCN channels towards more positive voltages.

While it is well established that the CNBD controls channel gating, it is frequently ignored that channel gating controls the CNBD. This mutual interaction is evident in patch clamp fluorometry experiments, where the movement of the voltage sensor changes the affinity of the CNBD for cAMP[8,9]. Due to the allosteric coupling of channel opening and ligand binding, the degree of ligand binding to the CNBD changes even under conditions of constant and basal levels of the second messenger[10]. The allosteric pathway between the CNBD and the VSD operates in both directions and involves the cytosolic C-linker and HCN domains and the S2-S3 and S4-S5 loops of the transmembrane domains[6,8,11].

In this work, we address another crucial aspect of the regulation of cAMP affinity in HCN channels, i.e. the large variability in affinity ($K_D$) and/or efficacy ($K_{1/2}$) values measured by isothermal titration calorimetry (ITC) and patch clamp, respectively. Of note, we adopt the definition by Colquhoun (1998)[12] where affinity is the rate constant for

[1]Department of Biosciences, University of Milan, Milano, Italy. [2]Department of Pharmacological and Biomolecular Sciences, University of Milan, Milano, Italy. [3]Institut für Physiologie II, Universitätsklinikum Jena, Jena, Germany. [4]Department of Neuroscience, Zuckerman Institute, Columbia University, New York, NY, USA. [5]Department of Biology, TU-Darmstadt, Darmstadt, Germany. [6]Institute of Biophysics Milan, Consiglio Nazionale delle Ricerche, Milano, Italy. [7]These authors contributed equally: Alessandro Porro, Andrea Saponaro. ✉e-mail: anna.moroni@unimi.it

binding to the inactive state while efficacy is the set of all other rate constants "which describe the transduction events that follow the initial binding reaction".

In isolated HCN2 fragments, the measured $K_D$ of the CNBD can vary from micromolar to nanomolar[5,13–15]. In these cases, discrepancies have been ascribed to the presence/absence of the C-linker and its role in forcing the tetramerization of the CNBD. Whatever the source of the variability, most relevant in this context is the experimental finding that the isolated CNBD can switch its affinity for cAMP by an order of magnitude. The same holds true when the efficacy ($K_{1/2}$) of cAMP-dependent gating is measured in different patch-clamp studies. An example is provided again by HCN2, where the $K_{1/2}$ for cAMP measured in inside-out macro patches from *Xenopus* oocytes is 100 nM[16–19]. Measurements performed in whole cell recordings in HEK 293 T cells, in contrast, provide micromolar concentrations for $K_{1/2}$ (this work). Further, in inside-out patch recordings from oocytes the $K_{1/2}$ value for the CNBD of HCN2 becomes micromolar when the C-term, downstream of the CNBD, is deleted[4]. These data confirm that, in HCN2, the affinity of the CNBD can convert from micromolar to nanomolar and suggest the presence of a control mechanism downstream of the CNBD.

A large variability in cAMP binding values occurs in HCN4 too. Also in this case, ITC measurements report variable affinity from micromolar to nanomolar[19–21]. While the latter results predict nanomolar $K_{1/2}$ values, such low affinities have never been reported from patch clamp studies of HCN4. In HEK cells, $K_{1/2}$ values are in the micromolar range, both in inside-out[22,23] and in whole-cell recordings[24] (and this work). In addition to this already complex scenario, it has to be mentioned that the $K_{1/2}$ values of HCN channels are cell-specific[25] since different cells express different protein interactors that control the affinity of the CNBD for cAMP[5,26–29].

Here we present evidence of a mechanism that controls cAMP affinity in HCN channels. Our study highlights that HCN channels (HCN1, HCN2 and HCN4) are endowed with an intramolecular mechanism that is formed by two folded alpha helices found downstream of the CNBD. These helices, named D and E (αDE) since they follow αC of the CNBD, are visible only in the cAMP-bound state of the HCN1 channel structure (Fig. 1a)[30,31]. This state dependency of αDEs is confirmed in the HCN4 structures where they are also visible in the presence of cAMP only, albeit at lower resolution than in HCN1[11]. In HCN1 (PDB 6UQF) αDEs clearly form a helix-turn-helix motif that effectively clamps αC from below. Their presence and potential modulatory function in HCN were so far overlooked, even though their sequence is highly conserved among subtypes (Fig. 1b).

In fact, previous studies of the isolated CNBD did not include these two helices, but only the first 4 residues of αD[4,13,14,19,32]. Notably, these four residues -SILL- form the initial turn of an alpha helix in the crystal structure of HCN2 CNBD (PDB 3U10).

Given the state-dependent appearance of the helices and their interaction with αC, that is part of the binding site of cAMP, here we tested if they control cAMP affinity in the full-length channel and in the isolated CNBD fragment. We discovered that αDEs indeed constitute an inherent control mechanism of CNBD affinity for cAMP. The CNBD can switch between two states with either nanomolar or micromolar affinity and this dramatic change depends on the interaction of both helices with αC. Our data further show that αDEs contribute to bind the exogenous β subunit TRIP8b, that similarly interacts with αC[5,6,28,33]. In summary, we found an endogenous mechanism that controls cAMP affinity in HCN and further showed that this mechanism is the target of exogenous regulation by interacting proteins.

## Results

The HCN1 structures solved in the presence of cAMP (PDB 5U6P, 6UQF) revealed the presence of two folded helices, named D and E (αDE), downstream of the C helix (αC) of the CNBD. Helices αDE, in

yellow in Fig. 1a, form a helix-turn-helix motif that contacts αC in the monomer (Fig. 1a blowup, gray). The inset shows that αDE surrounds αC from two sides, clamping the end part of it by means of two main interactions: a polar contact between D629 on αE and R593 on αC and an apolar contact between L601 and L602 on αD and I588 on αC. Supplementary Table 1 shows the complete list of putative interactions that we have identified between αDE and the CNBD. The amino acid sequence that codes for αDE is highly conserved among HCN subtypes (Fig. 1b) and the residues involved in the interactions shown in the inset are identical in hHCN1, hHCN2 and rbHCN4 (highlighted in gray or yellow, Fig. 1b), with the minor substitution of an isoleucine with a leucine in HCN4 (L709).

Given the crucial role of αC in cAMP affinity[4,5], we tested if αDEs control cAMP affinity in HCN and if this occurs in all subtypes. To this end, we introduced partial or full deletions of αDE in HCN4, HCN2 and HCN1 channels and analyzed their response to cAMP by patch clamp. In parallel, we measured the affinity for cAMP of the isolated CNBD fragments by Isothermal Titration Calorimetry (ITC).

### The region downstream of αDEs does not affect cAMP response in HCN4

Prior to investigating the role of αDE, we tested whether the C terminal portion downstream of the two helices contains any sequence that might affect cAMP affinity. To this end we introduced a stop codon in position R759 of rbHCN4, indicated by a gray arrowhead in Fig. 1b (ΔC-term). The latter was compared to rbHCN4 full-length (FL) in patch experiments performed in HEK293T cells. Figure 1c shows representative current records in response to voltage steps to −30 mV, −90mV and −150 mV (see detailed protocol in Material and Methods). Measurements were performed with and without 1 μM cAMP, a non-saturating concentration for HCN4[24] (Fig. 2c). We did not observe any significant difference between the two clones in kinetics, voltage-dependence, and response to cAMP.

Fitting the normalized conductance-voltage curves (G/Gmax)(V) to the Boltzmann function yielded nearly identical fitting parameters for the two clones: $V_{1/2}$ (in mV) was −101.3 ± 1.1 (FL) and −101.6 ± 2 (ΔC-term) in control conditions (solid lines) and −89.9 ± 1.6 (FL, shift of 11.4 mV) and −88.9 ± 1.5 (ΔC-term, shift of 12.7 mV) after addition of cAMP (broken lines). Thus, in our experimental conditions the C-terminal portion that follows αDE does not affect the cAMP response of rbHCN4.

### Deletion of αDEs reduces the cAMP response in HCN4

To study the role of αDE on channel function, we prepared three deletion mutants of rbHCN4 (Fig. 1b), shown in Fig. 2a: clone ΔE (aa 1-733) lacking αE; clone ΔDE (aa 1-718) lacking αDE; clone ΔDE' (aa 1-723) lacking αE and part of αD. The ΔDE' clone was tested because this deletion, which starts after the first four aa -SILL- of αD, has been extensively used in prior in vitro studies of the isolated C-linker-CNBD fragment[4,5,14,24,34].

Figure 2b shows, color-coded with the arrowheads of Fig. 1b, the mean activation curves of the constructs (Supplementary Fig. 1) without (full symbols, solid lines) and with 5 μM cAMP (empty symbols, broken lines), a non-saturating concentration in rbHCN4[24] (Fig. 2c). In basal conditions, the $V_{1/2}$ values of the FL and the mutants are almost identical (between −104 and −102 mV, Supplementary Table 2) indicating that the helices do not control the voltage-dependent response of the channel; on the contrary, the response of the channels to cAMP, evaluated as the shift in $V_{1/2}$, was either strongly reduced or abolished, depending on the degree of the deletion. In the FL the average shift in $V_{1/2}$ was 14.9 mV ± 2 mV and deletion of αE nearly halved the cAMP-induced shift (ΔE = 7.0 ± 1.9 mV). Deletion of both helices had an even higher effect since clone ΔDE essentially did not respond to cAMP (ΔDE = 1.4 ± 1 mV). Interestingly, the ΔDE' clone had a cAMP sensitivity like that of the ΔE clone (ΔDE' = 6.3 ± 1.9 mV), indicating that the

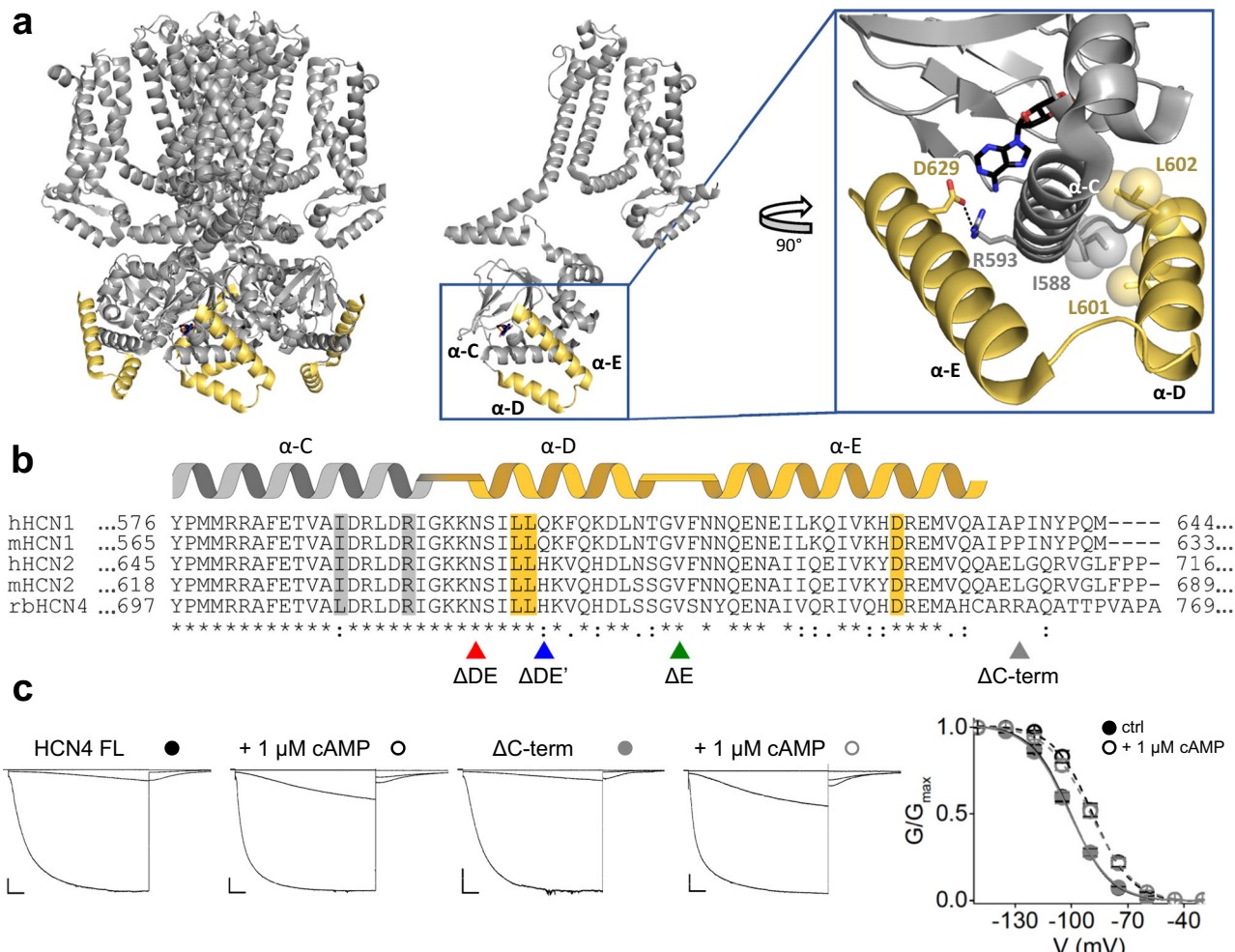

**Fig. 1 | Hydrophobic and hydrophilic interactions of the D and E helices with the C-helix. a** Ribbon representation of human HCN1 tetrameric structure bound to cAMP (PDB: 6UQF). Helices D and E of each monomer are shown in yellow and labeled. Boxed: Blow-up showing residues L601, L602 of D-helix and D629 of E-helix engaging hydrophobic and hydrophilic interactions with I588 and R593 on the C-helix of their own subunit. Numbering refers to hHCN1 sequence. **b** Multiple sequence alignment and secondary structure elements in the region of the D and E helices of human and mouse HCN1 (hHCN1 and mHCN1, Gene ID: 348980 and 15165, respectively), human and mouse HCN2 (hHCN2 and mHCN2, Gene ID: 610 and 15166, respectively) and rabbit HCN4 (rbHCN4, Gene ID: 100009452). Residues shown in the blow-up in panel a, are highlighted in gray and in yellow. Symbols below denote residue identity (*) and conservative (:) or semi-conservative (.) substitutions. Arrowheads indicate the last residue of the truncated constructs, color-coded as follow: ΔC-term gray, ΔE green, ΔDE' blue and ΔDE red. **c** Representative whole cell currents of HCN4 full length (HCN4 FL) and ΔC-term recorded at −30, −90 and −150 mV in control solution and with 1 μM cAMP in the patch pipette. Scale bars: 200 pA and 500 ms. Right: mean activation curves in control solution (full symbols) and with cAMP (empty symbols) from HCN4 FL (black) and ΔC-term (gray). Data are presented as mean ± SEM. Data fit with a Boltzmann function are plotted as solid (control) and dashed line (+1 μM cAMP). Calculated half-activation voltages ($V_{1/2}$) and inverse slope factors (k) are reported in Supplementary Table 2 together with the details on statistical analysis.

residues -SILL- of αD functionally complement the entire helix. Plotting the shift in $V_{1/2}$ as a function of cAMP concentration, we obtained the curves of Fig. 2c. In the FL, the half-maximal shift concentration $K_{1/2}$ was 1.35 μM, in agreement with previous reports[24]. Deletion of αE increased the $K_{1/2}$ value about 6-fold in ΔE (8.6 μM) and ΔDE' (9.3 μM). ΔDE, in contrast, showed a dramatic 28-fold increase in $K_{1/2}$, 38 μM.

While these results clearly highlight a crucial role of αDE in regulating the response of rbHCN4 to cAMP, two main aspects need to be investigated further: (i) Whether the helices affect the affinity of the CNBD for cAMP, or the efficacy of the ligand in facilitating channel gating; (ii) Whether αDE have a similar effect in other HCN subtypes.

We performed ITC experiments using a hHCN2 protein fragment that included the CNBD and the downstream sequences as in ΔDE, ΔDE', ΔE and ΔC-term constructs (see Material and Methods for details on the constructs). To specifically isolate the effect of helices DE on the modulation of cAMP affinity, all CNBD constructs started after the first three helices of the C-linker, thus ruling out any contribution of the latter[17,18].

All four hHCN2 CNBD constructs gave measurable saturating heat exchange curves upon addition of cAMP (Fig. 2d upper panels). Fitting of the binding isotherms with a single-site binding model (see "Methods") yielded the individual $K_D$ values for cAMP reported in Fig. 2d (lower panels), with mean group values shown in Fig. 2e and Supplementary Table 3. Deletion of αE and of part of αD, induced a 4-fold increase in $K_D$ (ΔC-term = 0.3 ± 0.01 μM, ΔE = 1.2 ± 0.1 μM, ΔDE' = 1.2 ± 0.03 μM). Deletion of αDE helices resulted in a much higher rise in $K_D$, about 30-fold (CNBD ΔDE = 9.5 ± 0.7 μM). The differences in affinity of the hHCN2 CNBDs mirrored the results obtained by electrophysiology with rbHCN4 channels, as well as mHCN2 where the $K_{1/2}$ measured in oocyte inside out patches is about 4–8-fold higher for the construct corresponding to ΔDE'[4] compared to wildtype values[12,13]. Therefore, we can conclude that αDE helices affect the affinity rather than the efficacy of cAMP binding and that the effect is conserved across HCN subtypes.

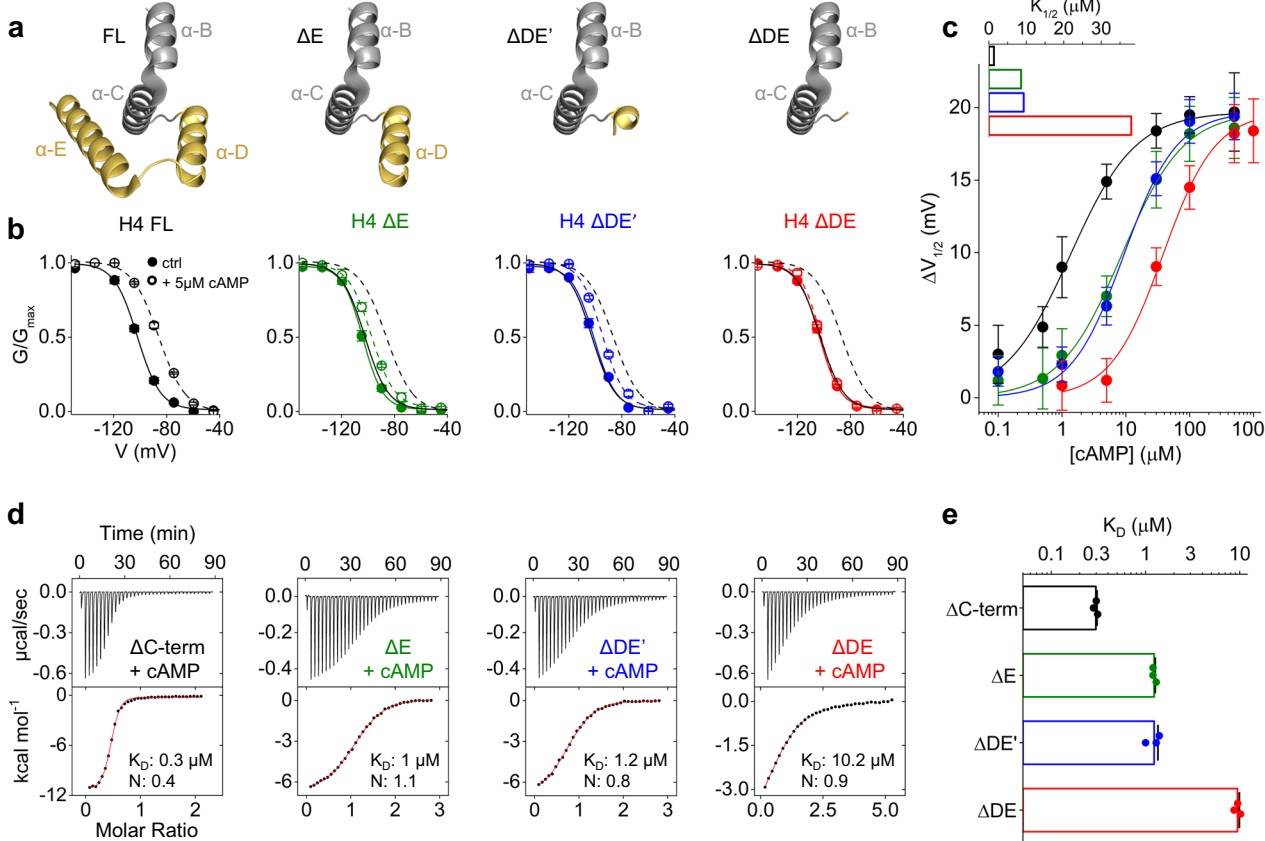

**Fig. 2 | Analysis of cAMP response and binding in HCN4 and HCN2 constructs deprived of the D and E helices. a** Ribbon representation of D and E helices, in yellow, shown with helices B and C of the CNBD, in gray, in the FL and in hypothetical representation of the truncated constructs ΔE, ΔDE' and ΔDE, with the portions removed (PDB:6UQF). **b** Mean activation curves measured in control solution (full symbols) and with 5 μM of cAMP (empty symbols) for HCN4 (H4) FL (black), ΔE (green), ΔDE' (blue) and ΔDE (red). Control (solid line) and 5 μM cAMP (dotted) line. FL lines are replotted, for comparison, in the other panels. Data are shown as mean ± SEM. Calculated half-activation voltages ($V_{1/2}$) and inverse slope factors (k) are reported in Supplementary Table 2 together with the details on statistical analysis. **c**, $\Delta V_{1/2}$ as a function of cAMP concentration for the constructs, color-coded as in panel b. Data fit to the Hill equation (solid lines) yielded half maximal effective concentration ($K_{1/2}$) values of 1.35, 8.6, 9.3, 38 μM and Hill coefficients (nH) values of 0.9, 0.9, 1, 1 for HCN4 FL, ΔE, ΔDE' and ΔDE constructs,

respectively. Each data point is an average of $n \geq 3$ experiments (exact numbers are provided in the source data file). Data are shown as mean ± SEM. **d** Representative binding curves of cAMP to purified human HCN2 CNBD constructs measured by ITC. CNBD ΔC-term includes helices DE, as shown in Fig. 1b. Top panels show heat changes (μcal/sec) following injections of cAMP into the chamber containing the protein. Bottom panels show the binding curve. The peaks were integrated, normalized to cAMP concentration, and plotted against the molar ratio (cAMP/CNBD). Solid red line represents a nonlinear least-squares fit to a single-site binding model yielding the equilibrium dissociation constant ($K_D$) and stoichiometry (N) values as shown for each representative sample. Mean $K_D$ and N values for the group are reported in Supplementary Table 3. **e** Mean $K_D$ values ± SEM ($n = 3$ experiments) from measurements shown in panel d, plotted on a logarithmic scale (see also Supplementary Table 3 for details on statistical analysis).

An interesting finding was that the binding stoichiometry (N) of the samples was close to 1 in the three mutants but only 0.4, in ΔC-term CNBD (Fig. 2d lower panels and Supplementary Table 3). A plausible explanation is that these higher affinity proteins retained cAMP during purification from *E. coli*. We validated this hypothesis by quantifying and comparing the cAMP content in ΔC-term and ΔDE proteins, from two different HCN subtypes, hHCN2 and mHCN1. The protein fragments were boiled to release in the supernatant the bound cAMP, which was then isolated with an ion exchange chromatography and quantified from its absorbance at λ = 254 nm. The data in Fig. 3 show that the two ΔC-term fragments indeed contained cAMP, and the calculated mean molar ratio [protein]:[cAMP] were 1.7 ± 0.2 (hHCN2) and 2.6 ± 0.2 (mHCN1). The ΔDE fragments, on the contrary, contained little cAMP bound, and their mean molar ratio were 30.2 ± 1 and 44.3 ± 2.4, for hHCN2 and mHCN1 respectively. These results support, by an independent approach, the interpretation that αDE strongly increase cAMP affinity of the CNBD. Moreover, by crossing electrophysiology and biochemical data, it emerges clearly that this mechanism operates in all HCN subtypes tested, HCN4, HCN2 and HCN1.

## How do αDEs affect the affinity of the CNBD?

Searching the hHCN1 structure for interactions that could underlie the effect of DE helices on CNBD affinity (Supplementary Table 1) we focused on the polar contact of αE and αC (D629 -R593) and the apolar contact of αD and αC (L601 and L602–I588) (inset, Fig.1a). Both interactions are at the C terminal part of αC, which is a known determinant of cAMP affinity. Indeed, this region of the helix folds upon cAMP binding and increases the affinity for the bound ligand[5,29].

It was previously reported that an alanine substitution of R635 in the full length mHCN2, equivalent to R593 in hHCN1, increased the $K_{1/2}$ value for cAMP (or reduced the potency) determined by fitting the data to a Hill equation. As for the role of the hydrophobic interaction in cAMP affinity, the comparison of ΔDE' and ΔDE has previously shown that the initial part of αD -SILL-, greatly increases affinity. We therefore introduced point mutations to test the effect of these interactions on cAMP affinity in FL rbHCN4 channels and in hHCN2 ΔC-term CNBDs.

## The salt bridge between αE and αC controls cAMP affinity

Figure 4a shows the salt bridge between αE and αC, that was further investigated by electrophysiology and ITC. Interestingly, the

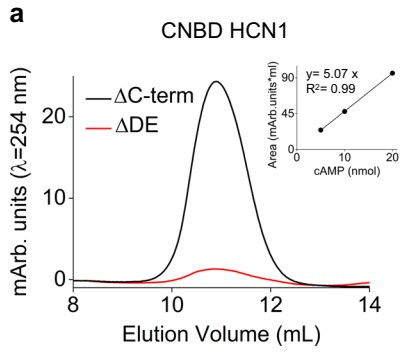

**Fig. 3 | Quantification of the cAMP content in the purified CNBDs of HCN1 and HCN2 subtypes.** Representative absorbance profiles, measured at λ = 254 nm, of cAMP molecules released by (**a**) HCN1 CNBD ΔC-term and CNBD ΔDE proteins (black and red lines, respectively) and (**b**) HCN2 CNBD ΔC-term and CNBD ΔDE proteins (black and red lines, respectively) after boiling. Mean molar ratio [protein]:[cAMP] ± SEM of 3 independent experiments are reported in the main text. Calibration curves are shown in the inset (Arb. units: arbitrary units).

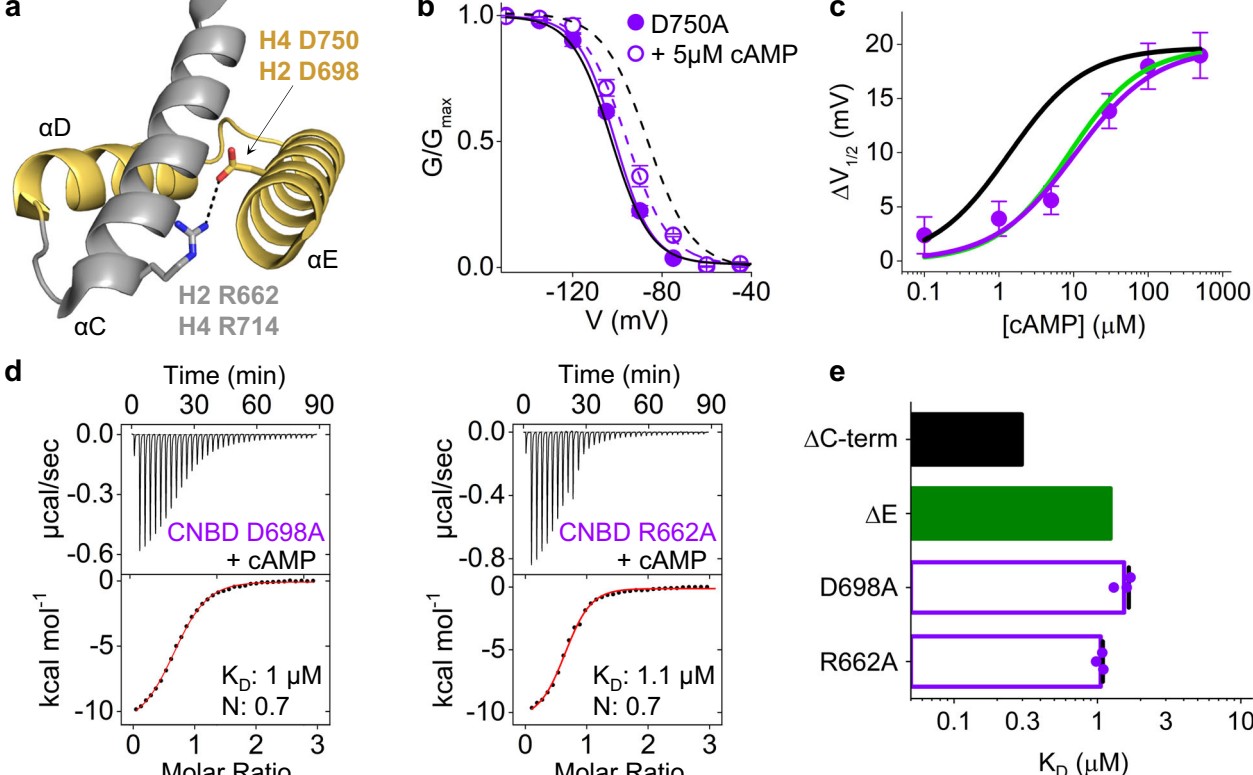

**Fig. 4 | Analysis of the salt bridge interaction between the E-helix and the C-helix. a** Ribbon representation of the D and E helices, in yellow, together with the C-helix of the CNBD, in gray (PDB:6UQF). The residues forming the salt bridge interaction (dotted line) are shown as sticks and labeled. Numbering refers to hHCN2 (H2) and rbHCN4 (H4) sequences. **b** Mean activation curves, measured by patch clamp, of HCN4 D750 (purple) in control solution (full symbol, solid line) and with 5 μM cAMP in the pipette solution (empty symbol, broken line). Activation curves of wild type HCN4 are plotted in black, without symbols, for comparison (replotted from FL in Fig. 2b). Half-activation voltage values ($V_{1/2}$) and inverse slope factors (k) are reported in Supplementary Table 2 together with the details on statistical analysis. Data are shown as mean ± SEM. Shift of $V_{1/2}$ as a function of cAMP concentration for HCN4 D750A construct (purple symbols). Each data point is an average of n ≥ 3 experiments (exact numbers are reported in the source data file). Data fit to the Hill equation (solid line) yielded half maximal effective concentration ($K_{1/2}$) of 10.4 μM and Hill coefficient (nH) of 0.8. Data are presented as mean ± SEM. Data for HCN4 FL (black line) and ΔE (green line) are replotted form Fig. 2c. **d** Representative ITC measurements of cAMP binding to purified hHCN2 CNBD D698A and R662A. Top panels show heat changes (μcal/sec) after cAMP injection into the chamber containing CNBD. Bottom panels show binding curves obtained from data displayed in the upper panel. The peaks were integrated, normalized to cAMP concentration, and plotted against the molar ratio (cAMP/CNBD). Solid red line represents a nonlinear least-squares fit to a single-site binding model yielding, in the present examples, equilibrium dissociation constant ($K_D$) and stoichiometry (N) values as shown. Mean $K_D$ and N values are reported in Supplementary Table 3 together with the statistical analysis. **e** Calculated $K_D$ values ± SEM for CNBD D698A and R662A (empty purple bars). Values for CNBD ΔC-term and ΔE are replotted as black and green bars without data points, from Fig. 2e.

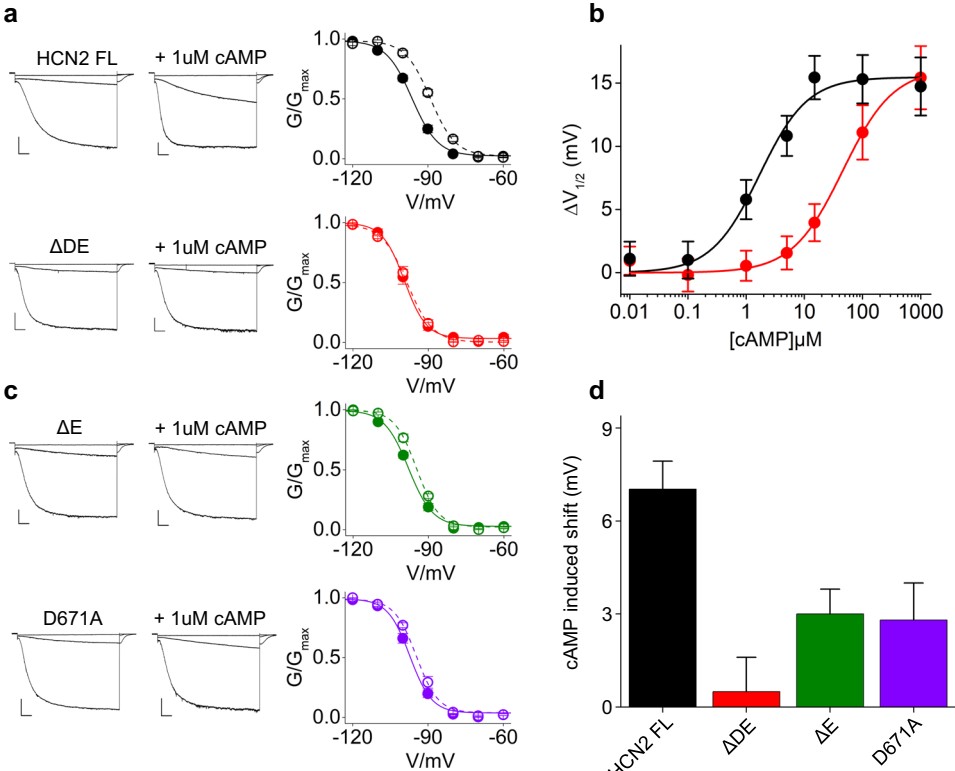

**Fig. 5 | Helices D and E control cAMP affinity in HCN2. a** Representative current traces (recorded at −30, −90 and −130 mV) and mean activation curves of HCN2 FL (black), ΔDE (red) in control solution (full symbols) and with 1 μM cAMP in the pipette solution (empty symbols). Scale bar is 200 pA x 500 ms. Half-activation voltage ($V_{1/2}$) and inverse slope factor (k) values are reported in Supplementary Table 2 together with the details on statistical analysis. Data are shown as mean ± SEM. **b** $\Delta V_{1/2}$ as a function of cAMP concentration for the constructs, color-coded as in panel a. Data fit to the Hill equation (solid lines) yielded half maximal effective concentration ($K_{1/2}$) values of 1.69 and 45.8 μM and Hill coefficients (nH) values of 1 and 1 for HCN2 FL and ΔDE constructs, respectively. Each data point is an average of $n \geq 3$ experiments (the exact number is reported in the source data file). Data are shown as mean ± SEM. **c** Representative current traces (recorded at −30, −90 and −130 mV) and mean activation curves of HCN2 ΔE (green) and D671A (purple) in control solution (full symbols) and with 1 μM cAMP in the pipette solution (empty symbols). Scale bar is 200 pA x 500 ms. Half-activation voltage ($V_{1/2}$) and inverse slope factor (k) values are reported in Supplementary Table 2 together with the details on statistical analysis. Data are shown as mean ± SEM. **d** cAMP-induced shift in $V_{1/2}$ in constructs from panels a and c. Data are shown as mean ± SEM.

neutralizing mutation D750A fully reproduced the effect of αE deletion (ΔE) in rbHCN4 currents, more than halving the shift in $V_{1/2}$ induced by 5 μM cAMP (Fig. 4b, Supplementary Table 2). Figure 4c shows a 7.7-fold increase in $K_{1/2}$ relative to the FL channel (black line, replotted from 2b) and overlaps with the ΔE curve (green line, replotted from 2b) confirming that this mutation recapitulates the deletion of αE.

The effect of disrupting the salt bridge between αE and αC was tested and confirmed by ITC in hHCN2 CNBDs. In this case, both salt bridge partners were individually mutated into alanine (D698A, R662A). Figure 4d shows the heat exchange peaks (upper panels) and the binding isotherms (lower panels) from which the affinities shown in Fig. 4e were calculated. The $K_D$ values for cAMP binding of D698A and R662A mutants are similar to each other (D698A = 1.5 ± 0.1 μM, R662A = 1 ± 0.03 μM) and to that measured for ΔE (replotted from Fig. 2e), confirming a fourfold reduction in affinity compared to the reference construct ΔC-term (replotted from Fig. 2e).

Altogether, the results highlight the role of the salt bridge as a key contact between αC and αE and indicate that this interaction controls cAMP efficacy in HCN4 as well as affinity in HCN2.

To complete the analysis, we further tested by electrophysiology the effect of αDE deletion on mHCN2.

In our experimental conditions, whole cell recordings in HEK 293 T cells, mHCN2 FL responded to 1 μM cAMP, with a shift of 7 ± 1.3 mV (Fig. 5a). The dose-response curve shown in Fig. 5b indicates that 1.69 μM is indeed the half saturating concentration ($K_{1/2}$) for this clone. This value is much higher than previously reported for mHCN2

measured in inside-out in oocytes[17,35] but coherent with a previous report of mHCN2 measured in whole cell in HEK293 cells[33]. The ΔDE mutant responded to 1 μM cAMP with no shift (Fig. 5a) and the calculated $K_{1/2}$ (Fig. 5b) was 45.8 μM, 27-fold higher than the wt. This result confirms that obtained in patch clamp with HCN4 and further confirms the results obtained in ITC with the mHCN2 CNBD (as fold-change not as absolute values).

As predicted, deletion of αE, or neutralization of the aspartate (D671A), did not affect the position of the activation curve in control solution but reduced the response to cAMP by 2.5-fold in both cases (ΔE = 3 ± 0.8, D671A = 2.8 ± 1.2) (Fig. 5c, d Supplementary Table 2).

The same experiment was repeated with hHCN1 (Fig. 6). In this case, the effect was compared to that of mutation R549E, that reduces cAMP affinity by 1000-fold[17]. In HEK 293 T cells, R549E mutant shows a left-shifted (−8.3 mV) $V_{1/2}$ (FL = −73.1 ± 0.4, R549E = −81.4 ± 0.9) (Fig. 6a, b). Similarly, deletion of αE (ΔE), or neutralization of the salt bridge partner aspartate (D671A in mHCN2, D629A in hHCN1), caused a left shift of about 4 mV in $V_{1/2}$ (ΔE = −78.6 ± 0.8, D629A = −77.5 ± 0.6), indicating a decrease in affinity (Fig. 6a–c).

Altogether, the results demonstrate that αE increases the affinity for cAMP in all tested HCN subtypes, and the salt bridge between αE and αC recapitulates the effect of αE on cAMP affinity.

**Apolar interaction between αC and αD**
An interesting finding related to αD is that the 4 initial residues -SILL-functionally substitute for the full-length helix. In other words, ΔDE'

and ΔE constructs show the same affinity drop (fourfold). Figure 7a shows that in the HCN1 structure, the two leucines of the -SILL-sequence form hydrophobic interactions with an isoleucine on αC, likely stabilizing the C terminal part of this helix. To test this hypothesis, we measured by ITC the affinity of the double mutant LL/GG in hHCN2.

The double mutant shows a tenfold increase in $K_D$ for cAMP binding (ΔC-term = 0.3 ± 0.01 μM; LL/GG = 3 ± 0.1 μM) (Fig. 7b, c and Supplementary Table 3), confirming the positive role of this apolar interaction in regulating the affinity of the wt CNBD for the ligand.

When we added mutation D698A to the double mutant to fully abolish the contact between αE and αC, the drop in the affinity of the triple mutant LL/GG-DA was increased to 23-fold, approaching the effect of ΔDE (Fig. 7b, c and Supplementary Table 3). Apparently, the double glycine substitution reduces, but does not completely disrupt, the positive contribution of αD to the folding and stability of αC.

### αC is a hub domain: comparison with TRIP8b

αC is a crucial determinant of cAMP affinity in the CNBD and undergoes large conformational changes upon ligand binding (Fig. 8a, left). It is, thus, not surprising that αDEs increase the affinity of the CNBD by interacting with αC and stabilizing it in its cAMP-bound state (holo) (Fig. 8a center). In line with the latter, TRIP8b, the β subunit of HCN channels that decreases their cAMP affinity, interacts with αC in the cAMP-unbound state (apo)[5,36,37]. Figure 8a, right, shows the structure of TRIP8b_nano, a peptide that recapitulates the effect of the full-length protein[29,33], in complex with the apo CNBD. Interestingly, both αE and TRIP8b_nano interact with and stabilize αC in its fully folded form. The insets further highlight that they form a salt bridge with the same arginine of αC (R593 in hHCN1 and R662 in hHCN2), raising the question of whether they compete for binding. In theory they should not, since TRIP8b_nano interacts with the apo CNBD, i.e., in the absence of αDE, while αE binds to the holo.

We have answered this question in two steps. First, we have verified by ITC whether the presence of the amino acid sequence of αDE may influence TRIP8b_nano binding to the apo CNBD. Figure 8b, c shows representative measurements from which mean $K_D$ values were obtained for TRIP8b_nano binding to the HCN2 CNBD constructs ΔC-term, ΔDE, ΔDE' and ΔE.

The affinity for ΔC-term and ΔE are similar (ΔC-term = $K_D$ = 0.6 ± 0.05 μM; ΔE = $K_D$ = 0.7 ± 0.01 μM), while removal of αD sequence (ΔDE) shows a threefold decrease ($K_D$ = 2.1 ± 0.07 μM) that is only partially recovered by the addition of the -SILL- residues, as in ΔDE' construct ($K_D$ = 1.2 ± 0.03 μM). This result indicates that the αD sequence contributes to the binding of TRIP8b_nano. The first four residues -SILL- presumably play the same role already highlighted for cAMP binding (Fig. 2), i.e. they stabilize αC in its folded conformation, which is needed for TRIP8b_nano binding. But the comparison between ΔDE and ΔDE' further shows that the second part of αD

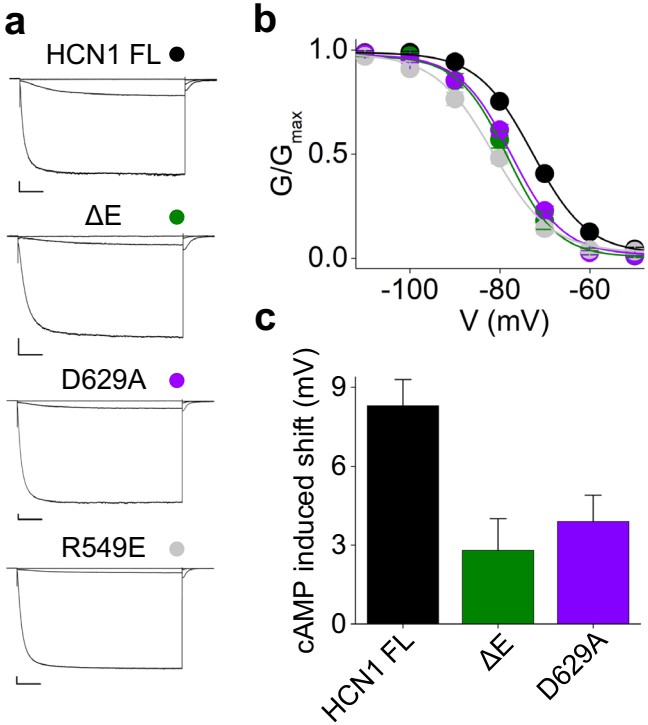

**Fig. 6 | helix E controls cAMP affinity in HCN1. a** Representative current traces (recorded at −30, −70 and −120 mV) of HCN1 FL (black), ΔE (green), D629A (purple) and R549E (gray). **b** Activation curves of HCN1 FL, ΔE, D629A and R549E; colors as in panel a. $V_{1/2}$ and k values are reported in Supplementary Table 2 together with the details on statistical analysis. Data are shown as mean ± SEM. **c** cAMP-induced shift in $V_{1/2}$ in constructs from a. For HCN1 constructs, the reference value is that of the mutant R549E, which is virtually cAMP-insensitive[17]. Data are presented as mean ± SEM.

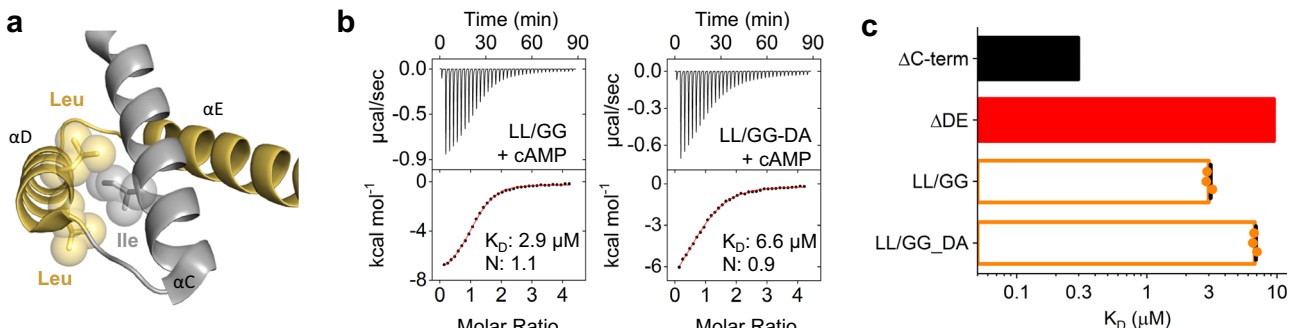

**Fig. 7 | ITC validation of the hydrophobic interactions between D-helix and C-helix. a** Structure of D and E helices (in yellow) along with the C-helix, shown in gray. Side chains of residues involved in the hydrophobic interactions are shown as sticks. Van der Waals surfaces are displayed as spheres (PDB:6UQF). **b** Examples of ITC thermogram obtained by titrating purified human HCN2 CNBD L670G/L671G with and without the mutation D698A in E-helix with cAMP (see Fig. 4). Top panels show heat changes (μcal/sec) during successive injections of cAMP. Bottom panels show binding curves obtained from data displayed in the upper panel. The peaks

were integrated, normalized to cAMP concentration, and plotted against the molar ratio (cAMP/CNBD). Solid red line represents a nonlinear least-squares fit to a single-site binding model yielding, in the present examples, equilibrium dissociation constant ($K_D$) and stoichiometry (N) values as shown. Mean $K_D$, N values and statistical analysis are reported in Supplementary Table 3. **c** Mean $K_D$ values ± SEM from measurements shown in panel B. Values for CNBD ΔC-term and ΔDE are replotted as black and red bars, without data points, from Fig. 2e.

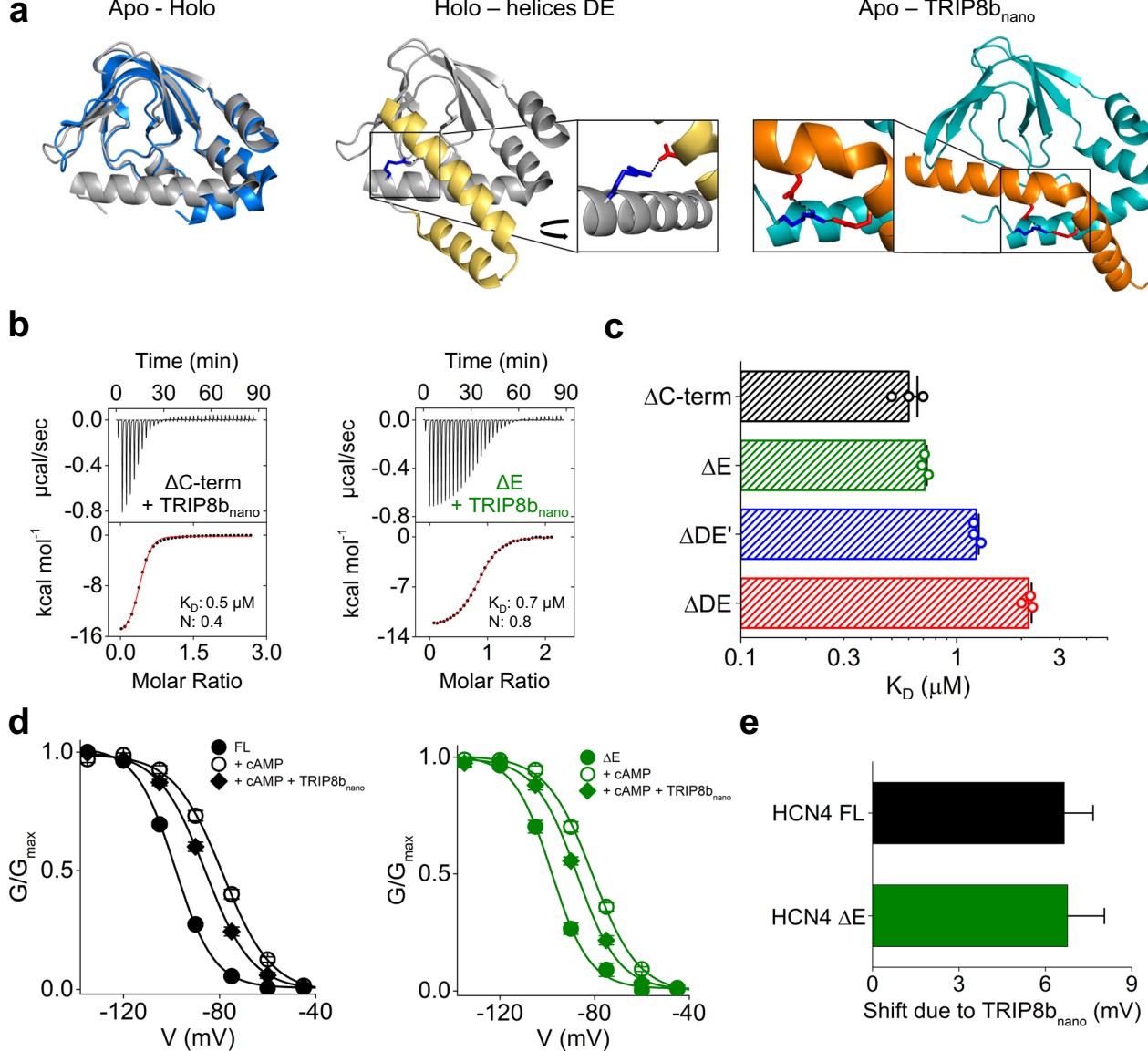

**Fig. 8 | Trip8b_nano binding to CNBD constructs with and without D and E helices. a** Left, superposition of the unbound (PDB: 5U6O, blue) and cAMP-bound (PDB: 6UQF, gray) structures of HCN2 CNBD; these constructs do not include helices DE. Center, structure of HCN1 (PDB:6UQF) highlighting the salt bridge, conserved in HCN subtypes, between Asp629 (red sticks) on helix E and Arg593 (blue sticks) on helix C. Right, structural model of HCN2 CNBD (light blue) in complex with TRIP8b_nano peptide (orange)[33]. Asp50 and 57 (red sticks) on the TRIP8b_nano peptide form salt bridges with Arg 662 (blue sticks) of C-helix. **b** Examples of ITC thermogram obtained by titrating purified human HCN2 CNBD ΔC-term, ΔE and with TRIP8b_nano peptide. Upper panel, heat changes (μcal/sec) during successive injections of TRIP8b peptide. Lower panel, binding curves obtained from data displayed in the upper panel. The peaks were integrated, normalized to TRIP8b peptide concentration and plotted against the molar ratio (TRIP8b_nano/CNBD). Solid red line represents a nonlinear least-squares fit to a single-site binding model yielding, in the present examples, equilibrium dissociation constant ($K_D$) and stoichiometry (N) values as shown. Mean $K_D$,N values and statistical analysis are reported in Supplementary Table 3. **c** Dissociation constant ($K_D$) values ± SEM of CNBD ΔC-term, ΔE, ΔDE' and ΔDE. **d** Mean activation curves of HCN4 FL (black) and ΔE (green) in control solution (full circles), with 15 μM and 60 μM cAMP for FL and ΔE respectively (empty circles), and with cAMP + 1 μM of purified TRIP8b_nano in the patch pipette (diamonds). Lines show Boltzmann fitting to the data. Half-activation voltage ($V_{1/2}$) and inverse slope factor (k) values are reported in Supplementary Table 4 together with the details on statistical analysis. Data are presented as mean ± SEM (where not visible, the error bars are within the symbol). **e** Mean shift due to TRIP8b_nano ± SEM calculated from data in (**d**).

sequence also contributes to form a binding site for TRIP8b_nano, which was previously unknown. Thus, we propose a model for TRIP8b binding to HCN channels that requires the folding of both αC and αD helices.

On the contrary, ΔE shows that in the apo CNBD the amino acid sequence of αE is neither necessary nor a hindrance for TRIP8b_nano binding. Note that, given the irrelevance of αE sequence for TRIP8b binding to the apo CNBD, we tend to exclude the hypothesis that αE folds in the presence of the regulatory protein. In conclusion, on the basis of the ITC experiments, we proceeded to test in patch

experiments HCN4 constructs possessing αD (FL and ΔE) in order not to alter TRIP8b affinity.

In the next step, we evaluated a potential interfering effect on TRIP8b_nano binding to αC exerted by a pre-folded αE, i.e., in the presence of cAMP (holo CNBD). To this end, we compared the effects of TRIP8b_nano on the FL and the ΔE constructs of HCN4 at concentrations of cAMP that varied to match the fourfold decrease in affinity of ΔE construct. Thus, we tested 15 μM cAMP and 60 μM, respectively, on FL and ΔE, that correspond to about 70% saturation (Fig. 2c). The tested concentration of TRIP8b_nano was kept constant, 1 μM, a non-saturating

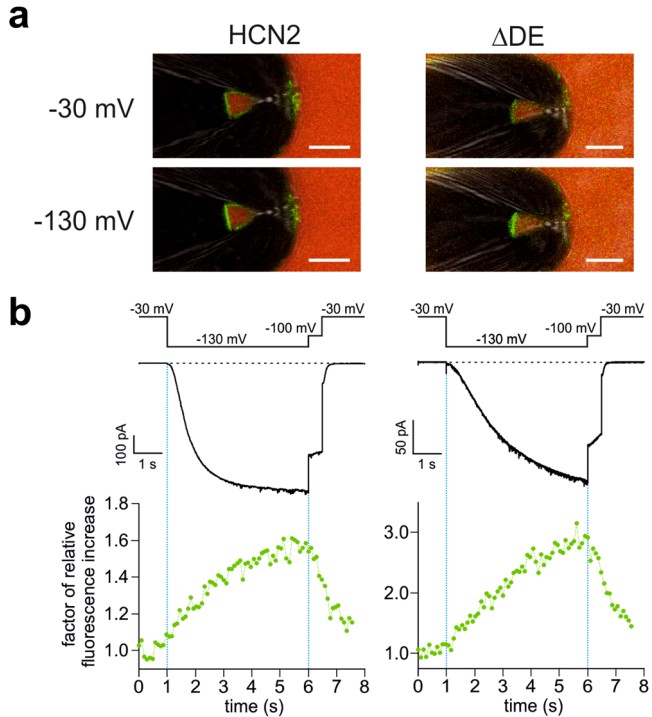

**Fig. 9 | Activation-induced affinity increase in HCN2 and ΔDE measured by confocal patch-clamp fluorometry (cPCF). a** Representative confocal images of pipette tips carrying excised macropatches from *X. laevis* oocytes. Either full-length mHCN2 (*left*) or ΔDE (*right*) channels were expressed. The green fluorescence signal of the patch is caused by binding of 0.25 μM f1cAMP to the binding sites of the functional channels. The red signal in the background is caused by the reference dye Dy647 (5 μM), used to subtract the background intensity of unbound f1cAMP. The scale bar is 10 μm. The microscope settings were similar throughout all wildtype and ΔDE recordings. For the sake of a better visibility of the ΔDE patch in exported images, the brightness of the green channel was increased during image processing, letting the original data unchanged. **b** Simultaneously measured current (black) and fluorescence traces (green) for full-length mHCN2 (*left*) and ΔDE (*right*). The voltage protocol is shown above. Fluorescence intensities at −30 mV were normalized to 1.

concentration at this cAMP levels[29]. Figure 8d, e and Supplementary Fig. 2a show that the effect of TRIP8b_nano, evaluated as a left shift of the activation curve due to the inhibitory effect of the regulatory protein on cAMP affinity, was the same in both constructs (FL = 6.7 ± 1.1 mV, ΔE = 6.8 ± 1.3 mV) (Supplementary Table 4). This indicates that αE does not affect the affinity of TRIP8b_nano for HCN4. Therefore, when CNBD is in the holo conformation, and thus αE is present, a competition between the latter and TRIP8b for the binding to αC seems excluded. It is worth noting that, compared to the αE - αC interaction, which is mediated by a single salt bridge (Fig. 4), TRIP8b binding to αC involves several residues of the helix, located both at its N and C terminus, and extends its contacts to other elements of the CNBD[7,33]. This large variety of contacts may explain why αE does not interfere with TRIP8b binding.

**Control on cAMP affinity exerted by the voltage sensor domain**
It was previously demonstrated that the affinity for cAMP of mHCN2 channels increases upon hyperpolarization-induced activation[8]. The findings presented here may raise the question whether the helices D and E are involved in this reciprocal communication between voltage sensor and CNBD.

We tested this hypothesis using confocal patch-clamp fluorometry (cPCF)[8], measuring current activation and ligand binding in parallel. We applied the fluorescently tagged cAMP derivative,

8-Cy3B-AHT-cAMP (f1cAMP)[28], to full length mHCN2 and ΔDE (mHCN2 1-640) mutant channels, both expressed in *Xenopus* oocytes.

First, we checked if the response of ΔDE (mHCN2 1-640) to cAMP in inside-out macropatches agreed with the data from HEK293T cells. The full-length mHCN2 channel responded to 10 μM cAMP, a saturating concentration, with a shift of about 20 mV. As expected, the ΔDE mutant showed a reduced shift of about 4 mV at the same cAMP concentration (Supplementary Fig. 3), paralleling the results in HEK293T cells and ITC.

Next, we measured the binding of f1cAMP to the channels in the resting and in the activated state, at −30 mV and −130 mV, respectively. As the binding affinity $BC_{50}$ for f1cAMP has been shown to be 0.25 μM for activated and 0.98 μM for non-activated mHCN2 channels[28], we chose to use a concentration of 0.25 μM. Figure 9a shows the pipette tip with excised macro patches from oocytes expressing mHCN2 and ΔDE channels. The green fluorescence signal of the patch is caused by binding of f1cAMP to the respective channels. Figure 9b shows the corresponding simultaneous recordings of time courses of current and fluorescence, which demonstrate the strong response of ΔDE channels to the application of the −130 mV voltage pulse. The mean increase of fluorescence intensity upon activation, $F_{-130 mV}/F_{-30mV}$, was 1.6 ± 0.1 (n = 9) for full length mHCN2 compared to 2.1 ± 0.3 (n = 4) for the ΔDE mutant.

In contrast, neither the full-length mHCN2 nor ΔDE channel showed any activation-induced binding increase at f1cAMP concentrations of 2.5 μM, most likely because this is a saturating concentration, in which all four binding sites are occupied most of the time. Thus, the mean fluorescence intensity at an activating voltage of −130 mV was similar to the fluorescence intensity at a non-activating voltage of −30 mV, namely $F_{-130 mV}/F_{-30mV} = 1.1 ± 0.04$ (n = 7) for full-length and 1.2 ± 0.06 (n = 3) for the ΔDE mutant.

Surprisingly, in these PCF experiments the deletion mutant ΔDE responded to f1cAMP at a similar concentration as the full-length mHCN2 channel. This suggests that for f1cAMP the affinity decrease due to the loss of helices D and E is not as pronounced as for unmodified cAMP. We speculate that the fluorophore moiety linked to position 8 of the cAMP molecule prevents this affinity decrease. This speculation fits with earlier results which show that several substitutions at position 8 of the purine ring, such as in 8-Br-cAMP or 8-AHT-cAMP or even more bulky groups as fluorophores, increase the affinity of cAMP and cGMP in isolated HCN CNBDs in the absence of the helices D and E[23].

In conclusion, these data show that helices D and E are not required for the reciprocal communication between voltage sensor and CNBD. The mechanisms underlying the activation-induced affinity increase are still intact in the absence of these two helices.

## Discussion
Previous work on HCN channels reported large discrepancies in cAMP affinity measured in the full-length protein or in the isolated CNBD. This raised the question whether the low affinity measured in the isolated CNBD is an experimental artifact, or rather reflects a physiologically relevant state of the protein. Here, we show that the CNBD of HCN channels is endowed with a previously unknown mechanism that controls its affinity for the ligand cAMP and allows the protein to be in a high or low affinity state. The key to this mechanistic control of cAMP affinity is represented by the αDE helices, which were first discovered in the high-resolution structure of HCN1, in the cAMP bound form. αDEs are located downstream of the CNBD where they form a helix-turn-helix motif that interacts with αC helix of the CNBD on both sides. Our data show that the interaction between αDE and the CNBD through αC is instrumental for defining the cAMP affinity not only of HCN1 but of HCN channels in general. This conclusion is supported by data showing that deletion of αDE in HCN4 drastically reduces the cAMP affinity by about 30-fold in patch experiments. Moreover, these

helices correspond to the regulatory element missing in the isolated CNBD fragments used so far in the biochemistry of HCN channels[5,7,13,14,33,36,37]. Indeed, the affinity for cAMP measured in functional studies with full-length channels and in binding assays with isolated CNBDs differ of about an order of magnitude (from micromolar to nanomolar). Our ITC data demonstrate that the addition of DE helices to the HCN2 CNBD fragment moves the affinity to the nanomolar range, mirroring patch clamp results.

The effect of both helices in this control mechanism seems to be roughly additive, in the sense that deletion of αE causes only a minor 4-fold decrease in affinity while affinity drops by a factor of 30 when αD, or more precisely only the first 4 residues (SILL) of this helix, are deleted. The assumption that this mechanism is a general feature of HCN channels is supported by results from ITC measurements with HCN2, but also from patch experiments performed on HCN2 and HCN1.

The combination of structure-guided mutations and functional assays suggest that αDEs specifically form a tweezer like clamp structure around the end part of αC, thus stabilizing the latter with several contacts. In this context, our data highlight the importance of a salt bridge between αE and αC; in addition, the SILL motive in αD interacts with αC. The presence or absence of these contact points fully recapitulates the changes in cAMP affinity observed in the presence or absence of αDE. These contacts are conserved in all isoforms, as shown by the identical result obtained with the neutralization of the salt bridge partner and /or exchange in HCN2 and HCN1, both in patch and in ITC experiments.

It is interesting to note that both crucial contacts of αD and αE are with amino acids at the end of αC. Based on structural studies, it is already known that this part of αC is very unstable and dynamic. We and others have previously shown that αC is partly unfolded in the cAMP-unbound CNBD and folds when the cyclic nucleotide binds to its pocket. Once folded, αC directly contacts the cAMP and acts as a lid, reducing cAMP unbinding from the pocket[38]. Collectively the structural and functional data advocate a scenario in which dynamic folding and unfolding of αDE can modulate cAMP affinity by stabilizing or destabilizing αC.

While this interpretation is mostly based on a simplified experimental system in which αDE are fully or partially deleted, the complementary data obtained with TRIP8b suggest that a modulation of cAMP affinity via stabilization of αC might also be physiologically relevant. The data show indeed that two opposite regulators of cAMP affinity in HCN channels, the endogenous mediated switch mechanism based on αDE and the exogenous switch mediated by the beta subunit TRIP8b, interact in a similar manner with αC to control cAMP affinity. Indeed, also the regulatory helix of TRIP8b, the so-called TRIP8b$_{nano}$ peptide[33], contains an aspartate residue that forms a salt bridge with the same arginine of αC that forms the contact with αE. In both cases, αC is in its fully folded conformation and in both cases the effect is that the affinity for cAMP changes. But, while αE stabilizes αC of the cAMP-bound CNBD, i.e., the high affinity state for the cyclic nucleotide, TRIP8b stabilizes αC of the cAMP-unbound CNBD, hence in the cyclic nucleotide low affinity state.

The finding that two independent mechanisms interact with αC to modulate cAMP affinity suggests a hub function of the CNBD for regulatory elements. Based on this finding, it will be worth investigating if other interactors, known to alter the HCN response to cAMP,[26,27] act on this helix as well.

Our data further highlight that αDE act independently of another known mechanism that controls the affinity of the CNBD, the voltage-dependent activation state of the channel[8]. Confocal patch clamp fluorometry (cPCF) measurements show that, unlike other components of the cytosolic machinery of HCN channels[39], the locking system of αDE is not required for the action of the voltage sensor to modulate cAMP affinity from the voltage sensor.

An interesting question is if cyclic nucleotide-gated (CNG) channels, which are also endowed with a CNBD domain, share a similar mechanism of control of ligand affinity, based on αDE. In the CNGA1/B1 heteromeric channel structure[40], the regulatory subunit CNGB1 contains a helix downstream of αC, termed D helix (αD). By binding to the 3-helix coiled-coil located below, it looks as if αD prevents the high affinity state of its own CNBD. Thus, in the presence of the ligand, αC adopts a cGMP-unbound conformation, resulting in an asymmetrically opened channel gate. In full agreement with our hypothesis, recent work has reported that Ca$^{2+}$/Calmodulin binds to αD of CNGB1, inducing conformational changes in the coiled-coil region that propagate Ca$^{2+}$/Calmodulin signaling from the cytosolic to the transmembrane region of the channels. The authors propose this mechanism as a key step in the Ca$^{2+}$/Calmodulin dependent reduction of the cyclic nucleotide sensitivity of CNG channels[41]. Intriguingly, the parallelism between HCN and CNG channels in the control of ligand affinity appears, therefore, not only at the level of the endogenous mechanism, the helices downstream of αC, but also at the level of the exogenous switch mediated by soluble protein interactors, TRIP8b for HCNs and Calmodulin for CNGs, both interacting with αD.

The main open question raised by our study remains the physiological role of the dynamic regulation of cAMP affinity, from nanomolar to micromolar, in HCN channels. In this respect, the already known regulation by TRIP8b strongly supports the view that a modulation of the affinity in a physiological context is a useful mechanism in response to the dynamic changes in cAMP concentrations in cells. It is presently accepted that, due to compartmentalization in subcellular nanodomains, the concentration of cAMP may not be uniform throughout a cell, but shows local hot spots and regions of low concentration[42]. Our results add another layer of complexity to this view in that the target protein can dynamically modulate its affinity for the signaling molecule. With such an endogenous system, which lowers the inherently high affinity of a binding site, the protein can adapt its sensitivity to the dynamic concentration range of a cell. While we cannot explain in absolute terms the changes in affinity and efficacy related to the helices, the identity of the fold changes induced by their removal/addition leaves no doubt that they are causally coupled.

## Methods

### Electrophysiology

**Constructs.** The cDNA encoding full-length human HCN1 was previously cloned into the pcDNA 3.1 (Invitrogen) mammalian expression vector. The cDNA encoding the full-length rabbit HCN4 was previously cloned in pCI (Promega) mammalian expression vector. The cDNA encoding the full-length mouse HCN2 was previously cloned in pCI (Promega) mammalian expression vector or in pGEMHE expression vector for oocytes. Single point mutations were introduced using the QuickChange XL-II kit (Agilent Technologies). Stop codons (TGA) were introduced into the plasmid DNA sequence sequences to create the following truncated constructs: rbHCN4 ΔC-term (1-758), rbHCN4 ΔE (1-733); rbHCN4 ΔDE' (1-723); rbHCN4 ΔDE (1-718); hHCN1 ΔE (1-612); mHCN2 ΔE (1-646); mHCN2 ΔDE (1-640). The introduction of a stop codon was considered equivalent to a deletion mutant because in the cases in which the two approaches were compared, we obtained the same results. The finding that key single mutations in the FL protein mimic the effect of a stop-codon mediated deletion further suggested to us that the rare event of a read through over a stop codon does not occur in our conditions. This is also consistent with the observation that the introduction of the stop codon always generated a single distinct phenotype, ruling out the possibility of a leaky read through phenotype. Finally, functional data obtained with the stop codon-induced deletions in the HEK293T cell environment induced the same fold-change in cAMP affinity measured using truncated proteins in ITC.

**HEK 293 T cells culture and transfection.** HEK293T cells were purchased from ATCC (Cat. #CRL-11268). Cells were periodically tested for Mycomplasma contamination using MycoAlter detection kit (Lonza) and resulted always negative. HEK293T cells were cultured in Dulbecco's modified Eagle's medium (Euroclone) supplemented with 10% fetal bovine serum (Euroclone), 1% Pen Strep (100 U/ml of penicillin and 100 µg/ml of streptomycin) and grown at 37 °C with 5% $CO_2$. When ~70% confluent, HEK293T cells were transiently transfected with cDNA using Turbofect transfection reagent (Thermo Fisher Scientific, Germany) according to the manufacturer's recommended protocol using 1 µg of the HCN gene-containing plasmid and 0.3 µg of EGFP-containing vector (pmaxGFP, Amaxa Biosystems).

**Patch clamp recordings in HEK cells.** 24 h after transfection cells were dispersed and single GFP$^+$ cells were selected for patch-clamp experiments at room temperature. Currents were recorded in whole-cell configuration either with a ePatch amplifier (Elements, Cesena, Italy) or with a Axopatch 200b amplifier (Molecular Devices); data acquired with the Axopatch 200b amplifier were digitized with an Axon Digidata 1550B (Molecular Devices) converter. Signals were acquired with a sampling rate of 5 kHz and low pass filtered at 2.5 kHz. Data analysis was performed using Clampfit 10.7 (Molecular devices). Patch pipettes were pulled from 1.5 mm O.D. and 0.86 mm I.D. borosilicate glass capillaries (Sutter, Novato, CA) and had resistances ranging from 3 to 6 MΩ. The pipettes were filled with a solution containing 10 mM NaCl, 130 mM KCl, 1 mM egtazic acid (EGTA), 0.5 mM $MgCl_2$, 2 mM ATP (magnesium salt), and 5 mM HEPES−KOH buffer (pH 7.2), while the extracellular bath solution contained 110 mM NaCl, 30 mM KCl, 1.8 mM $CaCl_2$, 0.5 mM $MgCl_2$, and 5 mM HEPES−KOH buffer (pH 7.4). To assess HCN channel activation curves, different voltage-clamp protocols were applied depending on the HCN subtype: for HCN1 holding potential was −20 mV (1 s), with steps from −30 mV to −120 mV (−10 mV increments, 3.5 s) and tail currents recorded at −40 mV (3.5 s); for HCN2, holding potential was −20 mV (1 s), with steps from −40 mV to −130 mV (−15 mV increments, 5 s) and tail currents recorded at −40 mV (5 s). For HCN4, holding potential was −20 mV (1 s), with steps from −30 mV to −150 mV (−15 mV increments, 4.5 s) and tail currents recorded at −40 mV (4.5 s). Adenosine 3′,5′-cyclic monophosphate (cAMP, SIGMA) was dissolved in MilliQ water to make a stock concentration of 100 mM and adjusting the pH to 7 with NaOH. Single-use aliquots were prepared and stored at −20 °C until the day of the experiment. From the stock solution, cAMP was added to the pipette solution to reach the desired final concentration by the mean of different intermediate dilutions. TRIP8b$_{nano}$ peptide was produced and purified as described in[33] and added to the pipette solution at 1 uM concentration.

Mean activation curves were obtained by fitting maximal tail current amplitude, plotted against the voltage step applied, with the Boltzmann equation: $y = 1/[1+\exp((V − V_{1/2})/k)]$, where V is voltage, y the fractional activation, $V_{1/2}$ the half-activation voltage, and k the inverse-slope factor $= RT/zF$ (all in mV). Mean $V_{1/2}$ values were obtained by fitting individual curves from each cell to the Boltzmann equation and then averaging all the obtained values. The shift in $V_{1/2}$ was plotted against the [cAMP] and fitted to a Hill equation ($Y = Y_{max}*(1/(1 + (K_{1/2}/x)^{nH}))$) to obtain $K_{1/2}$ the concentration for half-maximal shift. All fittings were performed using OriginPro software (OriginLab, Northampton, MA).

**Oocytes preparation.** Oocytes were surgically removed from adult females of the South African clawed frog *Xenopus laevis* under anesthesia (0.3% MS-222 (tricaine methanesulfonate) (Pharmaq, Fordingbridge, UK)). The surgery procedures were carried out in accordance with the German Animal Welfare Act with the approval of the Thuringian State Office for Consumer Protection

on 30.08.2013 and 09.05.2018. The oocytes were treated with collagenase A (3 mg/mL; Roche Diagnostics (Grenzach-Wyhlen, Germany)) for 105 min in $Ca^{2+}$-free Barth solution containing 82.5 mM NaCl, 2 mM KCl, 1 mM $MgCl_2$, and 5 mM HEPES (pH 7.5). Oocytes of stages IV and V were manually dissected and injected with cRNA-encoding mHCN2 channels of *Mus musculus* or constructs carrying a point mutation. After injection with cRNA, the oocytes were cultured at 18 °C for 2–10 days in Barth solution containing 84 mM NaCl, 1 mM KCl, 2.4 mM $NaHCO_3$, 0.82 mM $MgSO_4$, 0.41 mM $CaCl_2$, 0.33 mM $Ca(NO_3)_2$, 7.5 mM Tris, cefuroxime, and penicillin/streptomycin (pH 7.4). The procedures had approval from the authorized animal ethics committee of the Friedrich Schiller University Jena. The methods were performed according to approved guidelines. Oocytes harvested in our laboratory were complemented with ready-to-use oocytes purchased from EcoCyte Bioscience (Dortmund, Germany).

**Patch clamp fluorometry.** The binding of the fluorescently tagged cAMP derivative 8-Cy3B-AHT-cAMP (f1cAMP) and the ionic current were measured simultaneously in macropatches from *Xenopus laevis* oocytes by confocal patch-clamp fluorometry (cPCF) as described previously[8,36]. In 8-AHT-Cy3B-cAMP, a cAMP to which the fluorescent dye Cy3B (GE Healthcare, Frankfurt, Germany) was linked via an aminohexylthio spacer to position 8 of the adenosine moiety[43]. The recordings were performed with an LSM 710 confocal microscope (Carl Zeiss, Jena, Germany) and were triggered by the ISO3 software (MFK, Niedernhausen, Germany). To distinguish the fluorescence of the nonbound f1cAMP from that of the bound f1cAMP, a second, chemically related dye, DY647 (Dyomics, Jena, Germany), was also added to the bath solution. The 543- and 633-nm lines of an He-Ne laser were used to excite f1cAMP and DY647, respectively. For quantifying the bound f1cAMP, the fluorescence intensities of the red and the green channels were corrected for small offsets, and the fluorescence in the red channel was scaled to the fluorescence in the green channel in the bath. The difference between the measured green and the scaled red profile for each pixel of the confocal image represents the fraction of the fluorescence signal originating from the bound f1cAMP. The free patch membrane (patch dome) only was used to quantify binding by setting a mask at a region of interest. The fluorescence, F, was averaged over all pixels inside this mask and normalized in each patch concerning the fluorescence at saturating (f1cAMP) and full channel activation (−130 mV), $F_{max}$. The recording rate of the confocal images was 10 Hz.

### ITC and cAMP content assays
**Constructs.** The cDNA encoding residues 235–275 (TRIP8b$_{nano}$) of murine TRIP8b (splice variant 1a-4) was previously cloned into pET-52b (EMD Millipore) bacterial expression vector downstream of a Streptactin-binding (Strep) (II) tag sequence[33]. The cDNA encoding residues 521–672 of human HCN2 CNBD was previously cloned into a modified pET-24 bacterial expression vector downstream of a double His6-maltose binding protein (MBP) tag[5]. The cDNA encoding residues 441–592 of mouse HCN1 CNBD was previously cloned into a modified pET-24[13]. The longer version of human HCN2 (521-706) and mouse HCN1 (441-625) CNBDs containing D and E helices were also cloned into the modified pET-24 vector.

**Protein preparation.** HCN1/2 CNBD proteins, wt and mutants, and TRIP8b$_{nano}$ were expressed in *Escherichia coli* Rosetta strain (EMD Millipore) and purified following the procedure previously described[5] with minor modifications. Cells were grown at 37 °C in LB to an OD600 of 0.6 and induced with 0.4 mM isopropyl-1-thio-β-D-galactopyranoside overnight at 20 °C, and 3 h at 37 °C to express CNBD proteins and TRIP8b$_{nano}$, respectively. Cells expressing CNBDs were collected by centrifugation and resuspended in ice-cold lysis buffer [500 mM KCl,

30 mM Hepes (pH 7.4), 10% (wt/vol) glycerol] with the addition of 1 mM β-mercaptoethanol, 10 µg/mL DNase, 0.25 mg/mL lysozyme, 0.5 mM PMSF, and EDTA-free complete protease inhibitor cocktail (Roche) (1:1000). Cells were sonicated on ice 12 times for 20 s each time, and the lysate was cleared by centrifugation. CNBDs were purified by affinity chromatography on $Ni^{2+}$-nitrilotriacetic acid resin and eluted in lysis buffer plus 300 mM imidazole. The eluted proteins were then loaded into a HiLoad 16/60 Superdex 75 prep grade size exclusion column (SEC) (GE Healthcare), which was equilibrated with 150 mM KCl, 30 mM Hepes (pH 7.4), 0.5 mM PMSF, and EDTA-free complete protease inhibitor cocktail (Roche) (1:1000). PMSF and protease inhibitor cocktail were added to prevent degradation of the flexible helices D and E of HCN CNBDs. Cells expressing $TRIP8b_{nano}$ were collected by centrifugation and resuspended in ice-cold lysis buffer [150 mM NaCl, 100 mM Tris·Cl (pH 8), and 1 Mm EDTA] with the addition of 1 mM β-mercaptoethanol, 10 µg/Ml DNase, 0.25 mg/mL lysozyme, 0.5 mM PMSF, and EDTA-free complete protease inhibitor cocktail (Roche) (1:1000). Cells were sonicated on ice 12 times for 20 s each time, and the lysate was cleared by centrifugation. $TRIP8b_{nano}$ was purified by affinity chromatography using StrepTrap HP columns (GE Healthcare) and eluted in 150 mM KCl, 30 mM Hepes (pH 7.4), and 2.5 mM desthiobiotin. $TRIP8b_{nano}$ was then loaded into a HiLoad 16/60 Superdex 75 prep grade size exclusion column (SEC) (GE Healthcare), which was equilibrated with 150 mM KCl, 30 mM Hepes (pH 7.4), 0.5 mM PMSF, and EDTA-free complete protease inhibitor cocktail (Roche) (1:1000). CNBDs and $TRIP8_{nano}$ were run in the same SEC buffer to avoid heat changes due to the dilution of the titrant $TRIP8_{nano}$ into the titrated HCN2 CNBD proteins during ITC experiments (see the following chapter). All chromatographies were performed at 4 °C and monitored using the AKTApurifier UPC 10 fast protein liquid system (GE Healthcare).

**Isothermal titration calorimetry (ITC).** ITC measurements were performed at 25 °C using a MicroCal VP-ITC microcalorimeter (Malvern Panalytical, ITC) following the procedure previously described[5,33,44,45]. The $His_6$-MPB tagged HCN2 CNBD proteins (20 µM) were titrated with cAMP (200-500 µM in 150 mM KCl, 30 mM Hepes (pH 7.4), 0.5 mM PMSF, and 1:1000 EDTA-free complete protease inhibitor cocktail, same buffer of the titrated CNBDs), or with $TRIP8b_{nano}$ (200 µM) using injection volumes of 10 µL. Calorimetric data were analysed with MicroCal Origin software (version 7), and equations were described for the single-site binding model[38].

**cAMP content.** Determination of the cAMP content of purified CNBD proteins performed following the procedure previously described[13]. Endogenous cAMP bound to $His_6$-MBP tagged HCN1/2 CNBDs purified from *Escherichia coli* Rosetta strain (EMD Millipore) was released by boiling the protein sample (~1 mg) for 2 min. The boiled protein samples were centrifuged at maximum speed for 10 min, at room temperature, and the supernatant was diluted in 5 mM ammonium bicarbonate. The sample was loaded onto an anion exchange chromatography column (HiTrapQ (1 ml), GE Healthcare), previously equilibrated with 5 mM ammonium bicarbonate. After a washing step (7 column volumes) with 5 mM ammonium bicarbonate, cAMP was eluted with a linear gradient of ammonium bicarbonate (5–1000 mM) in 20 column volumes. Calibration curves were performed by loading in the HiTrapQ column 5, 10 and 20 nmoles of cAMP diluted in 5 mM ammonium bicarbonate. Chromatographies were performed at 4 °C and monitored using the AKTApurifier UPC 10 fast protein liquid system (GE Healthcare). Calculation of the area of the peaks of cAMP eluted from the HiTrapQ column was performed by using the software UNICORN5.11.

**Reporting summary**
Further information on research design is available in the Nature Portfolio Reporting Summary linked to this article.

## Data availability
The authors declare that the data supporting the findings of this study are available within the article and its supplementary information files, and from the corresponding author on request. The source data underlying Figures and Supplementary Figures are provided as a Source Data file. Previously published PDB codes are: 3U10; 5U6O; 5U6P; 6UQF. Source data are provided with this paper.

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

## Acknowledgements

We thank Federica Gasparri, Elena Foli for help with patch recordings and Alessandro Cogliati for protein purification; Fondazione Telethon grant GGP200221 (A.M.), Project Leducq Foundation for Cardiovascular Research "Fighting against sinus node dysfunction and associated arrhythmias (FANTASY)"—LFCR 219CVD03 (D.D. and A.M.), Cariplo Young Investigator grant no. 2018–0231 (A.S.) and NIH grant R01-NS109366 (B.S.) for financial support.

## Author contributions

A.M. conceived the study and with A.P. and A.S. designed the experiments. A.P., R.C., A.H.A., performed the patch clamp recordings in HEK cells, and U.E. and J.K. in oocytes; B.I., G.R. and A.S. performed protein purification and ITC experiments; A.P., A.S. and A.M. analyzed the results and wrote the manuscript. K.B., B.S., D.D., G.T. and A.M. revised the manuscript.

## Competing interests

The authors declare no competing interests.
