## [Peer Review File · Nature Communications]

A high affinity switch for cAMP in the HCN pacemaker channelsReviewers' Comments:

Reviewer #1:

Remarks to the Author:

I have attached a pdf of my review.

Summary and overall impression

Porro et al (2023) have carried out interesting experiments to better understand how cAMP binds to the HCN channel. According to the authors, the isolated C-linker and CNBD binds to cAMP 2-5 times less strongly than found for cAMP in patch clamp experiments. Therefore, they suggest that this region, in the context of the full channel, is modified such that cAMP binds to its site with greater affinity. The data suggest that the alpha helices D and E in the distal part of the CNBD, which are not present in experiments using the isolated CNBD, are required for stronger binding of cAMP to the channel. Without these helices in the intact channel the affinity for cAMP is also lower. Thus, helices D and E are required for high affinity binding of cAMP and its greater potency and, therefore, constitute part of a high affinity switch for this intracellular molecule. These helices interact with the C-helix to modify the binding site and enhance affinity. Finally, TRIP8b, an accessory subunit of HCN channels, may stabilize the lower affinity state by interaction with these regions.

Comments

I think the idea is very interesting. I think that, in general, measurement of currents, determination of activation curves and quantifying the effect of cAMP in combination with measurement of affinity by direct binding using ITC is excellent. Such a direct comparison between the EC₅₀ and separate binding experiments is challenging but, together, they yield better understanding of how cAMP binds and acts on the channel than does either measurement alone. Overall, I think that the idea that the C and E helix might enhance the binding of cAMP, and the EC₅₀ of cAMP, seems to be supported by the binding data and the electrophysiological data presented. This is an important feature because the ability of cAMP to bind and act on the channel determines its contribution to membrane potential and disruption of binding is associated with disease.

However, I found the paper a bit hard to follow and unclear for the reasons detailed below.

In Figure 1, the effect of cAMP at 1mM on FL HCN4 and delta C-term HCN4 are compared and are similar. I think that the assumption here is that the maximum effect of cAMP is the same for the full length and delta C-term HCN4? Otherwise, an experiment using a large concentration would be necessary? I also note that, in Figure 2, a concentration versus shift curve is presented for the full length HCN4 but not for the delta-C-term HCN4? Is it assumed that, for comparison with direct binding by ITC, the full-length and delta C-term respond the same way to cAMP as measured by patch clamp?

In line 387 methods, I note that, the rbHCN4 delta C-term channel was not detailed but there are measurements of current from this construct (Figure 1)?

A point raised here by the authors is that cAMP may be able to bind to the full-length channel more strongly than to the isolated C-linker and CNBD. Cyclic AMP is very potent in the full-length HCN2 channel where EC₅₀ values of about 100 nM have been determined by fitting the shift in V_{1/2} versus the concentration of cAMP with the Hill equation (e.g. Zhou et al 2007, Kusch et al, 2011).

The $V_{1/2}$ value itself is determined from a fit of the Boltzmann equation to the activation curve of the channel when hyperpolarization-activated currents are measured cells that express the particular HCN isoform. This is compared to binding values of cAMP to the isolated C-linker and CNBD which are much lower, in the neighborhood of 2- 4 micromolar (lines 53-61).

A series of papers is cited to support the values of binding affinity to the isolated C-linker and CNBD of between 2-4 micromolar. These include Chow et al (2012). However, in Chow et al, values of binding affinity were found to be of higher affinity (0.12 μM) and lower affinity (1.5 μM) when the protein was at higher concentrations ($\sim 100\mu\text{M}$ or greater) and where it formed predominantly tetramers. At lower concentrations, where tetramers were less frequent only a single lower affinity was found of approximately 1.5 μM . When the first two helices of the C-linker were removed and the protein was monomeric, only a single lower affinity was found of approximately 1.8 μM or so. It was proposed that interactions between the subunits, which include the C-linker and CNBD up to the C-helix, produced high affinity binding and, upon binding of one high affinity site, the remaining three sites were bound with lower affinity. This data suggests that the interactions between subunits may result in high affinity binding of cAMP.

It may be that, when the C-linker and CNBD forms part of the entire channel, the D and E helices that help to stabilize the binding site in such a high affinity structure?

It is mentioned that previous studies have found that the measured EC_{50} values are about 100nM for the HCN2 isoform. However, the data here using the full-length HCN4 isoform show that the EC_{50} is ten times larger, at 1.35 μM (legend in Figure 2). This EC_{50} value for HCN4 is similar to the values obtained in previous studies from your group e.g. 0.82 μM (Moller et al 2014), 1.53 μM (Milanesi et al, 2006) and 1.67 μM (Baruscotti et al, 2017). Previous studies have also found that cAMP is more potent in full-length HCN2 ($EC_{50} = 0.08 \mu\text{M}$) as opposed to full-length HCN2 with a C-linker and CNBD from HCN4 ($EC_{50} = 0.24 \mu\text{M}$) (Xu et al, JBC, 2010). What is responsible for the difference in EC_{50} values between HCN2 and HCN4? This difference makes the comparison between EC_{50} from patch clamp in HCN4 and binding to C-linker and CNBD in HCN2 more complicated. Also, there appears to be no EC_{50} value reported for HCN2 (or HCN1) here? There is a comment in this paper that 1 micromolar may be a half-saturating concentration here (line 200)?

The value for EC_{50} here is 1.35 μM for cAMP binding to the full-length HCN4, which is very close to the K_d values obtained by ITC for HCN2 delta E and delta DE', larger than the K_d value obtained for delta C-term ($\sim 0.3 \mu\text{M}$, I think), and smaller than the value for delta DE ($\sim 10 \mu\text{M}$). Comparing the values of EC_{50} with the K_d values obtained by ITC in micromolar;

1.35, 0.3 - 8.6, 1.2 - 9.3, 1.2 - 38, 10

So, it does not seem to be the case that the EC_{50} values (for HCN4) are lower than the K_d values obtained in the corresponding constructs (of HCN2) by ITC. Previous studies (e.g. Chow et al 2012, cited here) show that the K_d values for cAMP binding to the C-linker and CNBD (up to the C-helix) are similar to those found for HCN2. Again, this raises the issue of why the EC_{50} values for HCN4 are larger than those for HCN2.

However, the pattern of EC50 values obtained for the action of cAMP on the HCN4 constructs does parallel the pattern obtained for Kd values obtained for the binding of cAMP to the corresponding HCN2 constructs. This similarity does support the idea that the D and E helices promote a higher affinity binding of cAMP in both HCN2 and HCN4.

Why was HCN2 chosen for the ITC whereas the full concentration versus shift data was carried out using HCN4? Previous studies have been carried out with the purified HCN4 (Chow et al, 2012; Hayoz et al, 2017) by ITC while the full effect of cAMP has been carried out on both HCN1 and HCN2 (including papers from Prof. Kusch, Benndorf)?

In supplementary table three, there are Kd values listed that come from ITC experiments. There is a value for 'FL', 0.3 μ M but I do not see such a value in the bar graph in Figure 2e. Is this supposed to be delta C-term in the table or is it supposed to be FL in the bar graph?

It is a bit difficult to follow the various terms used to describe the concept of affinity in the manuscript. The following terms are used; K1/2, EC50, IC50, "half-saturating concentration" and Kd. My guess is that EC50, K1/2 and half-saturating concentration are in reference to a value derived from the Hill equation? It may be easier to follow if only one of these terms is used (and defined). Also, when defined in line 426, the Hill equation has another term, IC50.

"Dose-response relationships were obtained by fitting the cAMP induced shift in V1/2 against the [cAMP] with a Hill equation ($Y = Y_{max} * (1 / (1 + (IC50/x)^{nH}))$)."

Is it EC50 rather than IC50?

Kd is, presumably, the value obtained by ITC

A minor point here is that the relation is referred to as dose response but it may be more accurate to call this a concentration versus response or shift in V1/2 curve.

The larger issue here is how Kd is related to the dependence of the shift in the activation curve as a function of cAMP concentration and the EC50 value obtained by a fit of that relation to the Hill equation? The Hill equation probably does not represent the underlying mechanism as well as the allosteric models that describe HCN gating. According to allosteric models (including those proposed by Prof. Benndorf and Difrancesco), cAMP binds to the channel with some affinity (perhaps a Kd that is obtained by binding experiments) which then promotes an opening step. Cyclic AMP may then bind to the opening step with higher affinity to stabilize opening. The dependence of the shift in the activation curve as a function of cAMP concentration (and the EC50 value derived from a fit of that data to the Hill equation), depend on both binding steps and opening steps, and come from measurements of current. The Kd value comes from direct measurements of binding to the protein in solution. So, although the EC50 value is influenced by the Kd value(s), it may not represent the same thing and each may vary differently from the other. So, comparing the Kd directly with the EC50, and calling the latter value "affinity", seems confusing as opposed to defining it more clearly.

In the legend of Figure 2, it might be helpful to have the number of repetitions for the ITC experiments, which are 3 for each value and is found in the supplementary table.

In the legend of Figure 2 you have “Each data point is an average of $n > 3$ experiments”. Does this mean four experiments or greater?

In figure 2 it would be helpful to have a plot of the $K_{1/2}$ (EC50?) values determined for the fits of the concentration response curves to the Hill equation rather than only showing the values in the legend. These are important and I think having them in a bar graph would make it easier to directly compare them with the dissociation constants determined by ITC, which are already presented as bar graph.

In the bar graph presented in Figure 2e, it might be helpful to have more numbers along the K_d axis?

Supplementary Table 3. There is a typo in the description of the table, “dissociation constant”. Also in this table, “HCN2 CNBD (titrad)” – is this tetrad?

Line 109 – 1 micromolar is a non-saturating concentration for HCN4 according to a cited reference, Moller et al. In that paper, was the concentration versus shift curve determined in excised patches or in whole cell configuration? I think you could just refer to your figure 2c? I think that the experiments performed in this paper were only whole -cell as I did not see any methods for inside out experiments?

Line 125 – 5 micromolar is a “quasi saturating” concentration for rb HCN4. I am not sure this is true for either the concentration versus shift curve in Moller et al (cited), or in your figure 2c?

Figure 5. Line 200. Is 1 μ M cAMP a half- saturating concentration for this HCN2 clone? I thought that the half saturating was more like 100 nM (e.g. Zhou et al, 2007, other Siegelbaum papers, Kusch et al, Nature Chem Biol, 2011, other Benndorf papers)? Why was a full concentration versus shift curve not determined for HCN2 (or HCN1)? The shift in response to 1 micromolar cAMP seems very large for HCN1; is it possible that this is a maximal shift at the top of the concentration versus shift curve?

Even though the mutants follow your expectations, I think there may be uncertainty here without the full curve for HCN1 and HCN2, such as was carried out for HCN4 in Figure 2c. One reason is that the maximum effect of cAMP might be different for wild type and any of the mutants for HCN1 and HCN2. Another is that concentration versus shift curve may be steep in this range of concentrations. So, even small variability could mean large changes in the shift in response to one micromolar cAMP. Because, the effects seem smaller than for HCN4, this may mean that the difference could be due variability? Finally, I note that the values for HCN1 were examined a bit differently – were they subtracted first from R548E and then tested? (from supplementary table 2)?

The following is from the manuscript.

“In this case, we exploited the known finding that in HCN1 the $V_{1/2}$ is right-shifted in HEK293T cells, presumably because the channel is already saturated at the endogenous cAMP concentration of the cell line.”

What is this evidence for this? There is evidence that the activation curve of the HCN1 channel is depolarized as compared to the curve for HCN2 for other possible reasons. In Wainger et al Nature, 2002 (a relevant paper that is not cited here but could be?) the difference between HCN1 and HCN2 is proposed to arise from differential inhibition of opening by the CNBD up to the C-helix. They found that, in the absence of cAMP in excised patches, the activation curve is more depolarized for the HCN1 channel as compared to the HCN2 channel. However, when the CNBD up to the C-helix is removed, the position of the activation curve is depolarized for both channels to roughly the same voltages. The effect of removal is much stronger for HCN2 than for HCN1.

Are these experiments in Figure 5 carried out using the whole-cell patch configuration? And, if so, is the concentration not close to zero in the absence of added cAMP?

Legend for Figure 5 “For HCN1 constructs, the reference value is that of the mutant R539E, which is virtually cAMP insensitive”. In the table, you have 549E?

Line 177 “It was previously reported that mutation of R635 in mHCN2, equivalent to R593 in rbHCN4, affects $K_{1/2}$ of cyclic nucleotide binding.” I think it might be helpful to be more specific here e.g. an alanine substitution of this residue in the full-length channel increased the $K_{1/2}$ value (or reduced the potency) determined by fitting the data to a Hill equation.

Line 156 “An interesting finding was that the binding stoichiometry (N) of the samples was close to 1 in the three mutants but only 0.4, in Δ c-term CNBD”. It is suggested that the reason for this is because of cAMP than remains bound to the channel during the purification process. Is it possible that the stoichiometry is also influenced by misfolded protein during the purification process? Also, it might be easier to follow if a bit more detail is presented in this paper as opposed to having to search in the cited papers how purification was carried out. Is it possible that the difference in cAMP bound is due to some differences in growth of protein in bacteria and purification?

Line 166 “These results confirm, by an independent approach, that α DE strongly increase cAMP affinity of the CNBD.” These results support such an interpretation but I am not so sure they confirm it?

Figure 7e. Why was the delta C-term of HCN4 not used to examine the effect of TRIP8b as was the HCN2 delta C-term as for ITC?

More description of the isolated protein that is used for ITC might make it easier to follow how these measurements were carried out and interpreted. For example, what is the concentration of the protein used for ITC? In the paper, the authors indicate that this information is found in a previous publication (Saponaro et al, PNAS, 2014). In that publication, I think protein was used at 20 μ M and 15 μ M. Is this what was used here? The concentration of the protein is an important issue because larger concentrations may promote oligomerization and tetramers may bind to cAMP with higher affinity (Chow et al 2012). Some detail of how the binding heat was calculated and fit might also help to understand the meaning of stoichiometry? Overall, it might be better to include more details about the methods in general even if in the supplementary section.

Line 208 “tipically”, typically

Delta c-term or C-term?

In figure 8, a change in fluorescence of a cAMP analog is observed when the channels are activated by hyperpolarization. This is similar between the full-length and delta DE HCN4 channels. I know that these studies come from published papers from Prof Kusch and Benndorf but why is this change in fluorescence interpreted as an increase in affinity?

Reviewer #2:

Remarks to the Author:

This manuscript from the Moroni lab presents a comprehensive investigation into the structural mechanism underlying the modulation of the key helix αC within the cyclic nucleotide-binding domain (CNBD) of HCN channels. By using cryo-EM structures, the study reveals that the newly discovered helices αD - αE in proximity play a crucial role in this modulation. HCN channels are vital voltage-gated ion channels known for their involvement in the autonomous electrical activities of various cell types.

This new modulation mechanism, hitherto uncharacterized, relies on charge interactions and holds significant potential for determining the cAMP affinity of HCN channels. Through a combination of mutagenesis, Isothermal Titration Calorimetry (ITC), and patch-clamp fluorometry (PCF), the researchers demonstrate that the CNBD in full-length HCN channels undergoes regulation by other segments of the channel through direct interactions. By purifying the isolated CNBD in the absence versus in the presence of the D-E helices and performing binding assays, D-E helices were found to control the cAMP affinity of HCN channels.

The experimental procedures employed in this study are careful and thorough, providing robust support for the conclusions drawn. While the conceptual advancement resulting from these findings may not be groundbreaking, it does offer a solid new perspective on the modulation of HCN channels. Some changes especially in data interpretation could further improve the paper.

Questions and specific comments:

1. In the PCF experiments depicted in Fig. 8c, the channel with deleted D-E helices exhibited a similar fold change in fluorescence at a concentration of $0.5 \mu\text{M}$ f1-cAMP. However, it appears that the data of ΔDE for this concentration were missing. The representative figures in Fig. 8b indicated that the ΔDE construct displayed a larger fold change in fluorescence compared to the wild-type HCN2 (~ 3 versus ~ 1.6). Although this difference was not discussed and no statistical significance was detected, further experiments with a larger sample size (n) may be necessary to demonstrate reproducibility and potentially reveal a distinction at the $0.25 \mu\text{M}$ f1-cAMP concentration.
2. In line 296, "interaction between the VSD and the binding pocket for cyclic nucleotides is still intact". This conclusion needs to be modified. VSD and CNBD are affecting each other allosterically, not via direct interaction.
3. In the methods section outlining the Boltzmann equation for HCN channels, note that the fitting process is reversed compared to fitting the G-V relationship of depolarization-activated channels. Consequently, the slope factor "k" should not include the "-" sign in front of RT/zF .
4. In the methods, the authors used stop codons to generate deletions instead of simply deleting the sequence of interest. However, the introduced stop codons can sometimes lead to a read-through effect, which may vary in its degree depending on the surrounding DNA sequence. Therefore, it is worth mentioning whether the authors have taken this possibility into account.
5. In line 207, "presumably because the channel is already saturated at the endogenous cAMP concentration of the cell line". The reason that hHCN1 is more right-shifted than hHCN2 or hHCN4 is not well understood and could be due to other reasons, not because of the cAMP binding of hHCN1 is saturated. I suggest remove this explanation.

Minor points:

1. Line 269, "mV" was missing for ΔE .
2. Line 354, citation is in a different style.

3, Line 121, Clinker should be C-linker.

4, Line 208, typo of "typically"

Reviewer n.1

In Figure 1, the effect of cAMP at 1mM (please note: it is 1 μ M) on FL HCN4 and delta C-term HCN4 are compared and are similar. I think that the assumption here is that the maximum effect of cAMP is the same for the full length and delta C-term HCN4? Otherwise, an experiment using a large concentration would be necessary? I also note that, in Figure 2, a concentration versus shift curve is presented for the full length HCN4 but not for the delta-C-term HCN4? Is it assumed that, for comparison with direct binding by ITC, the full-length and delta C-term respond the same way to cAMP as measured by patch clamp?

Yes, the assumption is indeed that the maximum effect of cAMP is the same for rbHCN4FL and DC-term, since the maximum effect of cAMP does not change for any of the shorter mutants either (Fig 2c). We chose 1 μ M cAMP because at this concentration any increases or decreases in the response can be easily detected in whole cell patch clamp recordings. As indicated in the manuscript, we compared the behavior of the rbHCN4FL and DC-term constructs to rule out any effects of sequences downstream of helices DE in modulating cAMP affinity. Since we did not find any effects, we are able to compare the patch experiments, where the control is FL, with ITC experiments, where the control is delta C-term (which of course also lacks the portion upstream of the CNBD, see below).

In line 387 methods, I note that, the rbHCN4 delta C-term channel was not detailed but there are measurements of current from this construct (Figure 1)?

Thank you for pointing it out, we have now added this information.

A point raised here by the authors is that cAMP may be able to bind to the full-length channel more strongly than to the isolated C-linker and CNBD. Cyclic AMP is very potent in the full-length HCN2 channel where EC50 values of about 100 nM have been determined by fitting the shift in $V_{1/2}$ versus the concentration of cAMP with the Hill equation (e.g. Zhou et al 2007, Kusch et al, 2011)

The $V_{1/2}$ value itself is determined from a fit of the Boltzmann equation to the activation curve of the channel when hyperpolarization-activated currents are measured cells that express the particular HCN isoform. This is compared to binding values of cAMP to the isolated C-linker and CNBD which are much lower, in the neighborhood of 2- 4 micromolar (lines 53-61).

A series of papers is cited to support the values of binding affinity to the isolated C-linker and CNBD of between 2-4 micromolar. These include Chow et al (2012). However, in Chow et al, values of binding affinity were found to be of higher affinity (0.12 μ M) and lower affinity (1.5 μ M) when the protein was at higher concentrations (\sim 100 μ M or greater) and where it formed predominantly tetramers. At lower concentrations, where tetramers were less frequent only a single lower affinity was found of approximately 1.5 μ M. When the first two helices of the C-linker were removed and the protein was monomeric, only a single lower affinity was found of approximately 1.8 μ M or so. It was proposed that interactions between the subunits, which include the C-linker and CNBD up to the C-helix, produced high affinity binding and, upon binding of one high affinity site, the remaining three sites were bound with lower affinity. This data suggests that the interactions between subunits may result in high affinity binding of cAMP.

It may be that, when the C-linker and CNBD forms part of the entire channel, the D and E helices that help to stabilize the binding site in such a high affinity structure?

The reviewer raises a very good point. It is indeed confusing to include studies on constructs that include or do not include the C-linker in the same set of citations, as we are aware of, and we fully agree that, the C-linker affects the affinity of the CNBD for cAMP. Chow et al (2012) clearly demonstrated this, and also Lolicato et al, 2011 came to the same conclusion. It is also possible, as the reviewer suggests, that in the context of the full-length channel the DE helices help stabilize the CNBD in a high affinity conformation promoted by the presence of the C-linker.

In the present manuscript, we intentionally chose to work with a fragment containing the CNBD only, without the C-linker, in order to isolate the effects of the DE helices from the known effects of the C-linker on cAMP affinity.

We have revised the manuscript to better articulate these points upfront by modifying the text and also separating the citations referring to *in vitro* binding studies employing CNBD constructs which include or exclude the C-linker (page 3, lines 61-63).

It is mentioned that previous studies have found that the measured EC50 values are about 100nM for the HCN2 isoform. However, the data here using the full-length HCN4 isoform show that the EC50 is ten times larger, at 1.35 uM (legend in Figure 2). This EC50 value for HCN4 is similar to the values obtained in previous studies from your group e.g. 0.82 uM (Moller et al 2014), 1.53 uM (Milanesi et al, 2006) and 1.67uM (Baruscotti et al, 2017). Previous studies have also found that cAMP is more potent in full-length HCN2 (EC50 = 0.08 uM) as opposed to full-length HCN2 with a C-linker and CNBD from HCN4 (EC50 = 0.24 uM) (Xu et al, JBC, 2010). What is responsible for the difference in EC50 values between HCN2 and HCN4? This difference makes the comparison between EC50 from patch clamp in HCN4 and binding to C-linker and CNBD in HCN2 more complicated.

We agree with the Reviewer, it is presently unclear what determines the different affinity for cAMP in HCN4 and HCN2, or HCN1. An additional difficulty is posed by the observation that $K_{1/2}$ values may differ even for the same isoform when tested in different cellular context (e.g. inside-out patches from *Xenopus* oocytes vs whole cell recordings from HEK293 cells, see also answer below). Why the values differ between methods is not fully understood at present. For these reasons, we believe a direct comparison of absolute values between patch clamp and ITC experiments is not informative. We could have performed the patch and the ITC experiments on the same subtype, rbHCN4, but even in this case we wouldn't have obtained comparable values in absolute terms because the ITC constructs are much simpler and cannot contain all the elements that contribute to determine the EC50 in the FL protein (C-linker, HCND, TM domain and so on). At the same time, we wanted to study more than one subtype to determine whether the effect of helices DE is a general property of HCN4, 2, and 1, where their sequence is conserved (not in HCN3). Therefore, we chose to study the effect of DE helices on rbHCN4 in patch clamp and on hHCN2 in ITC. Given that their affinities/EC50 are very different in terms of absolute values, we always compare fold-changes in affinity rather than absolute values. In this manner, we can work with different subtypes showing that the role of DE helices is the same in all subtypes.

Also, there appears to be no EC50 value reported for HCN2 (or HCN1) here? There is a comment in this paper that 1 micromolar may be a half-saturating concentration here (line 200)?

We work in whole-cell patch clamp from HEK293 cells and we have determined (but not published) the dose-response curve obtained by plotting the $V_{1/2}$ shift vs cAMP concentration (in the pipette) for mHCN2. This value differs from the published $K_{1/2}$ for mHCN2, which was obtained in inside-out patches from *Xenopus* oocytes. As noted above, it is currently unknown why the $K_{1/2}$ obtained for HCN2 and HCN4 in recordings from HEK293 cells are generally higher than those obtained for the same constructs in inside out patches from *Xenopus* oocytes (e.g. Xu et al. JBC 2010 vs Milanesi et al 2006). This is why we indicate in the text, line 200, “In our experimental conditions...” (meaning whole cell), 1 μ M cAMP gives a shift of 7 mV, half of the maximal shift (14-15 mV) obtained with this clone at saturation (15 μ M) (Porro et al, Elife. 2019 Nov 26;8:e49672. doi: 10.7554/eLife.49672).

As for HCN1, we don't need to determine the EC50 because we perform the experiment without adding cAMP. We use as a read out, the left-shift induced in HCN1 (in HEK 293 cells) by whichever means that reduce the affinity for cAMP of the CNBD. This can be for instance induced by the addition of TRIP8b_{nano} peptide (Porro et al, 2018), or mutations (shown here is RE in the CNBD), or, as in this case, deletion of the helices.

The value for EC50 here is 1.35 μ M for cAMP binding to the full-length HCN4, which is very close to the Kd values obtained by ITC for HCN2 delta E and delta DE', larger than the Kd value obtained for delta C-term (~ 0.3 μ M, I think), and smaller than the value for delta DE (~10 μ M). Comparing the values of EC50 with the Kd values obtained by ITC in micromolar;

1.35, 0.3 - 8.6, 1.2 - 9.3, 1.2 - 38, 10

So, it does not seem to be the case that the EC50 values (for HCN4) are lower than the Kd values obtained in the corresponding constructs (of HCN2) by ITC. Previous studies (e.g. Chow et al 2012, cited here) show that the Kd values for cAMP binding to the C-linker and CNBD (up to the C-helix) are similar to those found for HCN2. Again, this raises the issue of why the EC50 values for HCN4 are larger than those for HCN2.

However, the pattern of EC50 values obtained for the action of cAMP on the HCN4 constructs does parallel the pattern obtained for Kd values obtained for the binding of cAMP to the corresponding HCN2 constructs. This similarity does support the idea that the D and E helices promote a higher affinity binding of cAMP in both HCN2 and HCN4.

As detailed in our replies above, we agree with the Reviewer that it is very confusing to directly compare absolute values of EC50 across different methodologies in vivo and with results from ITC studies in vitro. Why the values differ between cellular contexts or methods (whole cell vs inside out) and between HCN subtypes is a very good question but beyond the scope of this study.

In terms of in vitro studies, as discussed above, we cannot compare prior ITC values obtained in the presence of the C-linker and CNBD, with our values obtained in ITC with the CNBD only, because the C-linker can change the affinity of the CNBD. Also, we cannot directly compare the absolute values of EC50 of HCN4 obtained on FL proteins by patch clamp (in whole cell) with Kd of hHCN2 obtained by ITC on isolated fragments. In conclusion, we can only compare the fold changes in EC50 and in K_D (what the Reviewer defines here as “pattern”) induced by the partial or full deletion of the helices DE. And, as the Reviewed indeed noticed, the fold changes are basically identical.

Why was HCN2 chosen for the ITC whereas the full concentration versus shift data was carried out using HCN4? Previous studies have been carried out with the purified HCN4 (Chow et al, 2012; Hayoz et al, 2017) by ITC while the full effect of cAMP has been carried out on both HCN1 and HCN2 (including papers from Prof. Kusch, Benndorf)?

We chose to work with HCN2 in ITC because we have previously determined the K_D for this protein fragment in our laboratory (Saponaro et al, 2014) and we used HCN4 in patch clamp because we wanted to confirm that the effect of helices DE was conserved also in other subtypes. As discussed above, we reasoned that direct comparisons of absolute $K_{1/2}$ values in patch clamp and K_d values in ITC are not informative, and so we only ever compare fold-changes in these parameters.

In supplementary table three, there are K_d values listed that come from ITC experiments. There is a value for 'FL', 0.3 μ M but I do not see such a value in the bar graph in Figure 2e. Is this supposed to be delta C-term in the table or is it supposed to be FL in the bar graph?

Thanks for pointing it out, we have corrected it in Figure 2e. The construct name is Δ C-term and FL was a mistake.

It is a bit difficult to follow the various terms used to describe the concept of affinity in the manuscript. The following terms are used; $K_{1/2}$, EC_{50} , IC_{50} , "half-saturating concentration" and K_d . My guess is that EC_{50} , $K_{1/2}$ and half-saturating concentration are in reference to a value derived from the Hill equation? It may be easier to follow if only one of these terms is used (and defined). Also, when defined in line 426, the Hill equation has another term, IC_{50} .

*"Dose-response relationships were obtained by fitting the cAMP induced shift in $V_{1/2}$ against the [cAMP] with a Hill equation ($Y = Y_{max} * (1 / (1 + (IC_{50}/x)^n)$))."*

Is it EC_{50} rather than IC_{50} ?

The Reviewer is right. IC_{50} in Hill equation was wrong and we changed in EC_{50} . Moreover, we decided to use $K_{1/2}$ throughout the manuscript, instead of EC_{50} .

K_d is, presumably, the value obtained by ITC

Yes.

A minor point here is that the relation is referred to as dose response but it may be more accurate to call this a concentration versus response or shift in $V_{1/2}$ curve.

We have removed the dose-response term from the manuscript.

The larger issue here is how K_d is related to the dependence of the shift in the activation curve as a function of cAMP concentration and the EC_{50} value obtained by a fit of that relation to the Hill equation? The Hill equation probably does not represent the underlying mechanism as well as the allosteric models that describe HCN gating. According to allosteric models (including those proposed by Prof. Benndorf and Difrancesco), cAMP binds to the channel with some affinity (perhaps a K_d that is obtained by binding experiments) which then promotes an opening step. Cyclic AMP may then bind to the opening step with higher affinity to stabilize opening. The

dependence of the shift in the activation curve as a function of cAMP concentration (and the EC50 value derived from a fit of that data to the Hill equation), depend on both binding steps and opening steps, and come from measurements of current. The Kd value comes from direct measurements of binding to the protein in solution. So, although the EC50 value is influenced by the Kd value(s), it may not represent the same thing and each may vary differently from the other. So, comparing the Kd directly with the EC50, and calling the latter value “affinity”, seems confusing as opposed to defining it more clearly.

We agree on the above, and in our opinion a way to summarize the whole point is that by patch clamp we measure “efficacy” of cAMP, while in ITC, we measure binding, i.e. “affinity” of cAMP to the CNBD.

There is no way to directly compare the two values. Nonetheless, the remarkable finding of our present study is that the fold-change induced by the helices in both values is the same. This, in our opinion, strongly suggests that what we measure in patch clamp is the effect of helices DE on “affinity”, even if the absolute values do not match the ITC ones, for all the reasons discussed above (presence of C-linker, etc etc).

In the legend of Figure 2, it might be helpful to have the number of repetitions for the ITC experiments, which are 3 for each value and is found in the supplementary table.

This is a good suggestion, and we have added this information.

In the legend of Figure 2 you have “Each data point is an average of $n > 3$ experiments”. Does this mean four experiments or greater?

It was a mistake, we substituted it with $n \geq 3$.

In figure 2 it would be helpful to have a plot of the $K_{1/2}$ (EC50?) values determined for the fits of the concentration response curves to the Hill equation rather than only showing the values in the legend. These are important and I think having them in a bar graph would make it easier to directly compare them with the dissociation constants determined by ITC, which are already presented as bar graph.

We have added the inset to panel b of the new Figure 2.

In the bar graph presented in Figure 2e, it might be helpful to have more numbers along the Kd axis?

We have plotted in a log scale values of 0.1, 0.3, 1, 3, 10 in all ITC histograms (new Panels in Figures 2e, 4e, 6c, 7c).

Supplementary Table 3. There is a typo in the description of the table, “dissociation constant”.

Thank you for pointing this out, this was corrected.

Also in this table, “HCN2 CNBD (titrad)” – is this tetrad?

We apologize for the typing error, it is “titrated”.

Line 109 – 1 micromolar is a non-saturating concentration for HCN4 according to a cited

reference, Moller et al. In that paper, was the concentration versus shift curve determined in excised patches or in whole cell configuration? I think you could just refer to your figure 2c? I think that the experiments performed in this paper were only whole-cell as I did not see any methods for inside out experiments?

Yes, all the patch clamp experiments are in whole-cell and we now refer to Figure 2c, thank you for the suggestion.

Line 125 – 5 micromolar is a “quasi saturating” concentration for rb HCN4. I am not sure this is true for either the concentration versus shift curve in Moller et al (cited), or in your figure 2c?

The reviewer is right, we have changed it into “ a non-saturating”.

Figure 5. Line 200. Is 1 μM cAMP a half- saturating concentration for this HCN2 clone? I thought that the half saturating was more like 100 nM (e.g. Zhou et al, 2007, other Siegelbaum papers, Kusch et al, Nature Chem Biol, 2011, other Benndorf papers)? Why was a full concentration versus shift curve not determined for HCN2 (or HCN1)?

The discrepancy is related to the fact that we are measuring mHCN2 in HEK293 cells in the whole-cell configuration (Porro et al 2019) rather than in inside-out patches from Xenopus oocytes (Zhou et al, 2007, other Siegelbaum papers, Kusch et al, Nature Chem Biol, 2011, other Benndorf papers).

As shown in Porro et al 2019, in whole cell recordings from HEK293 cells for mHCN2 1 μM cAMP yields a $V_{1/2}$ shift that is approximately half of the maximal $V_{1/2}$ shift obtained in the presence of 10 μM cAMP. Here, we chose to work at concentrations far from saturation to better appreciate the shift induced by DE.

The shift in response to 1 micromolar cAMP seems very large for HCN1; is it possible that this is a maximal shift at the top of the concentration versus shift curve?

Please note that in this experiment, with HCN1, we don't add cAMP to the pipette, but we simply measure the left shift induced by the decrease in affinity (see next three replies below).

Even though the mutants follow your expectations, I think there may be uncertainty here without the full curve for HCN1 and HCN2, such as was carried out for HCN4 in Figure 2c. One reason is that the maximum effect of cAMP might be different for wild type and any of the mutants for HCN1 and HCN2. Another is that concentration versus shift curve may be steep in this range of concentrations. So, even small variability could mean large changes in the shift in response to one micromolar cAMP. Because, the effects seem smaller than for HCN4, this may mean that the difference could be due variability? Finally, I note that the values for HCN1 were examined a bit differently – were they subtracted first from R548E and then tested? (from supplementary table 2)?

The experiments performed with HCN2 and HCN1 are qualitative, they intend to show that the helices exert the same kind of effect shown in HCN4, HCN2 and HCN1.

Concerning the experiment with HCN1, yes, the reference channel is the mutant R548E. We have added new text to the manuscript to better explain the rationale of this experiment (page 8, lines 217-225) and also in the answer to the next question below.

The following is from the manuscript.

“In this case, we exploited the known finding that in HCN1 the $V_{1/2}$ is right-shifted in HEK293T cells, presumably because the channel is already saturated at the endogenous cAMP concentration of the cell line.”

What is this evidence for this? There is evidence that the activation curve of the HCN1 channel is depolarized as compared to the curve for HCN2 for other possible reasons. In Wainger et al Nature, 2002 (a relevant paper that is not cited here but could be?) the difference between HCN1 and HCN2 is proposed to arise from differential inhibition of opening by the CNBD up to the C-helix. They found that, in the absence of cAMP in excised patches, the activation curve is more depolarized for the HCN1 channel as compared to the HCN2 channel. However, when the CNBD up to the C-helix is removed, the position of the activation curve is depolarized for both channels to roughly the same voltages. The effect of removal is much stronger for HCN2 than for HCN1.

We agree with this observation, raised also by Reviewer 2, that the reasons why the HCN1 $V_{1/2}$ is right-shifted are not fully understood. Here we exploit the known finding that HCN1 is pre-bound to cAMP when expressed in HEK293 cells (or in *E.coli*, Lolicato et al 2011) and that mutations like R549E (this study and Porro et al, 2019) or addition of peptides like TRIP8b_{nano} (Saponaro et al, 2018), induce a left shift by reducing CNBD affinity for cAMP. We have modified the sentence as follow:

“The same experiment was repeated with hHCN1 (Figure 5b). In this case, we exploited the known finding that in HEK293 cells a reduction in cAMP affinity typically shifts the HCN1 $V_{1/2}$ value to the left. In this case, the mutation R549E, known to reduce cAMP affinity by 1000-fold¹³, induced a -8 mV shift (FL= -73.1 ± 0.4 mV, R549E= -81.4 ± 0.9 mV) (Fig. 5b,c). Similarly, deletion of α E (Δ E), or neutralization of the salt bridge partner aspartate (D671A in mHCN2, D629A in hHCN1), caused a left shift of about 5 mV in $V_{1/2}$ (Δ E= -78.6 ± 0.8 mV, D629A= -77.5 ± 0.6 mV).”

Are these experiments in Figure 5 carried out using the whole-cell patch configuration? And, if so, is the concentration not close to zero in the absence of added cAMP?

Exactly, indeed HCN1 has some cAMP bound, as shown by the left shift induced for instance by either the introduction of mutation, or the addition TRIP8b_{nano}, or the removal of the DE helices. All these three conditions reduce the affinity of the CNBD to cAMP.

Legend for Figure 5 “For HCN1 constructs, the reference value is that of the mutant R539E, which is virtually cAMP insensitive”. In the table, you have 549E?

The right number is R549E, thank you.

Line 177 “It was previously reported that mutation of R635 in mHCN2, equivalent to R593 in rbHCN4, affects $K_{1/2}$ of cyclic nucleotide binding.” I think it might be helpful to be more specific here e.g. an alanine substitution of this residue in the full-length channel increased the $K_{1/2}$ value (or reduced the potency) determined by fitting the data to a Hill equation.

We have introduced this change in the text.

Line 156 “An interesting finding was that the binding stoichiometry (N) of the samples was close to 1 in the three mutants but only 0.4, in Δc -term CNBD”. It is suggested that the reason for this is because of cAMP than remains bound to the channel during the purification process. Is it possible that the stoichiometry is also influenced by misfolded protein during the purification process?

We show in Figure 3, that the content of cAMP in the protein matches the changes in N . See also Lolicato et al, 2011, where we use the same approach.

Also, it might be easier to follow if a bit more detail is presented in this paper as opposed to having to search in the cited papers how purification was carried out.

Sorry about it, we have now added the detailed purification protocol.

Is it possible that the difference in cAMP bound is due to some differences in growth of protein in bacteria and purification?

We grow the clones in parallel, and we have already demonstrated that the CNBD of HCN1 contains more cAMP than those of the other subtypes and that this is related to the affinity (Lolicato et al, 2011).

Line 166 “These results confirm, by an independent approach, that α DE strongly increase cAMP affinity of the CNBD.” These results support such an interpretation but I am not so sure they confirm it?

The Reviewer is correct and we have changed the text accordingly.

Figure 7e. Why was the delta C-term of HCN4 not used to examine the effect of TRIP8b as was the HCN2 delta C-term as for ITC?

As controls, we have used in all patch clamp experiments the FL clone and in ITC the ΔC -term construct that contains the CNBD and the helices DE. We have verified in patch that FL and ΔC -term behave the same, because we need to use this truncation in ITC.

More description of the isolated protein that is used for ITC might make it easier to follow how these measurements were carried out and interpreted. For example, what is the concentration of the protein used for ITC? In the paper, the authors indicate that this information is found in a previous publication (Saponaro et al, PNAS, 2014). In that publication, I think protein was used at 20 μ M and 15 μ M. Is this what was used here? The concentration of the protein is an important issue because larger concentrations may promote oligomerization and tetramers may bind to cAMP with higher affinity (Chow et al 2012).

The Reviewer is totally right, sorry for this. We have added the protein concentration in our revised section of the Materials and Methods, as well as in Suppl. Table 3. Both the absence of C-linker helices A' and B' in our constructs and the use of 20 μ M protein concentration in our ITC experiments rule out the occurrence of oligomerization.

Some detail of how the binding heat was calculated and fit might also help to understand the meaning of stoichiometry? Overall, it might be better to include more details about the methods

in general even if in the supplementary section.

We have now added all the details for protein purification and ITC measurements in the Material and Methods and in Suppl. Table 3.

Line 208 “typically”, typically

Thank you, we corrected the spelling.

Delta c-term or C-term?

Delta C-term, we have corrected it throughout the text to match the figure labeling.

In figure 8, a change in fluorescence of a cAMP analog is observed when the channels are activated by hyperpolarization. This is similar between the full-length and delta DEHCN4 channels. I know that these studies come from published papers from Prof Kusch and Benndorf but why is this change in fluorescence interpreted as an increase in affinity?

As the Reviewer noted, the experiments presented in Figure 8 follow an experimental approach initially published by Kusch and co-workers in 2010 (Neuron 67, 75–85, July 15, DOI 10.1016/j.neuron.2010.05.022), which demonstrated an increase in cAMP binding affinity for HCN2 channels upon channel activation. As described therein and in the present manuscript, an increase in fluorescence intensity is observed during the course of a hyperpolarizing voltage pulse. Because the concentration of applied fluorescently tagged cAMP is kept constant during the voltage pulse, the increase in fluorescence intensity at the membrane patch must result from an increase in binding affinity. A false interpretation due to flcAMP binding to membrane lipids or endogenous membrane proteins could be ruled out by performing the appropriate controls (Kusch et al 2010, Neuron 67, 75–85, July 15, DOI 10.1016/j.neuron.2010.05.022).

Reviewer #2

1. In the PCF experiments depicted in Fig. 8c, the channel with deleted D-E helices exhibited a similar fold change in fluorescence at a concentration of 0.5 μ M fl-cAMP. However, it appears that the data of ΔDE for this concentration were missing. The representative figures in Fig. 8b indicated that the ΔDE construct displayed a larger fold change in fluorescence compared to the wild-type HCN2 (~3 versus ~1.6). Although this difference was not discussed and no statistical significance was detected, further experiments with a larger sample size (n) may be necessary to demonstrate reproducibility and potentially reveal a distinction at the 0.25 μ M fl-cAMP concentration.

The reviewer is correct in pointing out that the representative example data in Figure 8b might indicate a higher degree of affinity increase for the mutant than for the wildtype channel. However, these higher values were obtained only for two of the samples tested, and indeed the grouped data at 0.25 μ M flcAMP indicate no statistically significant difference.

The purpose of the experiments presented in Figure 8 was to test if there is still an activation-induced increase in binding affinity after deleting helices D and E. This is clearly demonstrated at a concentration of 0.25 μ M flcAMP, since all mutant samples tested consistently showed an increase

in fluorescence. Because these data already satisfyingly answer our question, in the revised manuscript we decided to remove data points for other f1cAMP concentrations from the figure. As we now point out in the revised text, modifications at position 8 of the cAMP molecule might themselves affect the binding affinity to a certain degree, which would require a detailed analysis to separate effects of the ligand from effects of helices D and E. This would be beyond the scope of this study.

To adapt the figure panels so as to more adequately convey the message of this experiment and avoid confusion, we decided (1) to skip panel 8c and (2) present the summary data obtained at 0.25 μ M and 2.5 μ M f1cAMP as numbers in the text. Moreover, we added a supplementary figure S3 to show the steady state activation curves with and without cAMP for mHCN2 and Δ DE as obtained in *Xenopus* oocytes.

We revised the whole paragraph concerning the PCF experiments, which now reads as follows:

“It was previously demonstrated that the affinity for cAMP of mHCN2 channels increases upon hyperpolarization-induced activation (Kusch et al., 2010). The findings presented here may raise the question whether the helices D and E are involved in this reciprocal communication between voltage sensor and CNBD.

We tested this hypothesis using confocal patch-clamp fluorometry (cPCF) (Kusch et al., 2010), measuring current activation and ligand binding in parallel. We applied the fluorescently tagged cAMP derivative, 8-Cy3B-AHT-cAMP (f1cAMP) (Otte et al., 2019), to full length mHCN2 and Δ DE (mHCN2 1-640) mutant channels, both expressed in *Xenopus* oocytes.

First, we checked if the response of Δ DE (mHCN2 1-640) to cAMP in inside-out macropatches agreed with the data from HEK293T cells. The mHCN2 channel responded to 10 μ M cAMP, a saturating concentration, with a shift of about 20 mV. As expected, the Δ DE mutant showed a reduced shift of about 4 mV at the same cAMP concentration (Supp. Fig. 3), paralleling the results in HEK 293 cells and ITC.

Next, we measured the binding of f1cAMP to the channels in the resting and in the activated state, at -30 mV and -130 mV, respectively. As the binding affinity BC_{50} for f1cAMP has been shown to be 0.25 μ M for activated and 0.98 μ M for non-activated mHCN2 channels (Otte et al., 2019), we chose to use a concentration of 0.25 μ M. Figure 8a shows the pipette tip with excised macropatches from oocytes expressing mHCN2 and Δ DE channels. The green fluorescence signal of the patch is caused by binding of f1cAMP to the respective channels. Figure 8b shows the corresponding simultaneous recordings of time courses of current and fluorescence, which demonstrate the strong response of Δ DE channels to the application of the 130 mV voltage pulse. The mean increase of fluorescence intensity upon activation, $F_{-130\text{ mV}}/F_{-30\text{ mV}}$, was 1.6 ± 0.1 (n=9) for full length mHCN2 compared to 2.1 ± 0.3 (n=4) for the Δ DE mutant.

In contrast, neither the mHCN2 wt nor Δ DE channel showed any activation-induced binding increase at f1cAMP concentrations of 2.5 μ M, most likely because this is a saturating concentration, in which all four binding sites are occupied most of the time. Thus, the mean fluorescence intensity at an activating voltage of -130 mV was similar to the fluorescence intensity at a non-activating voltage of -30 mV, namely $F_{-130\text{ mV}}/F_{-30\text{ mV}} = 1.1 \pm 0.04$ (n=7) for wt and 1.2 ± 0.06 (n=3) for the Δ DE mutant.

Surprisingly, in these PCF experiments the deletion mutant Δ DE responded to f1cAMP at a similar concentration as the wildtype mHCN2 channel. This suggests that for f1cAMP the affinity decrease due to the loss of helices D and E is not as pronounced as for unmodified cAMP. We speculate that the fluorophore moiety linked to position 8 of the cAMP molecule prevents this affinity decrease. This speculation fits with earlier results which show that several substitutions at position 8 of the purine ring, such as in 8-Br-cAMP or 8-AHT-cAMP or even more bulky groups as fluorophores, increase the affinity of cAMP and cGMP in isolated HCN CNBDs in the absence

of the helices D and E (Möller et al., 2014).

In conclusion, these data show that helices D and E are not required for the reciprocal communication between voltage sensor and CNBD. The mechanisms underlying the activation-induced affinity increase are still intact in the absence of these two helices.”

2. In line 296, “interaction between the VSD and the binding pocket for cyclic nucleotides is still intact”. This conclusion needs to be modified. VSD and CNBD are affecting each other allosterically, not via direct interaction.

The Reviewer is right. We have modified the whole paragraph, removing the sentence among other corrections.

3. In the methods section outlining the Boltzmann equation for HCN channels, note that the fitting process is reversed compared to fitting the G - V relationship of depolarization-activated channels. Consequently, the slope factor “ k ” should not include the “-” sign in front of RT/zF .

The Reviewer is right, thank you for pointing this out. We have deleted the “-”.

4. In the methods, the authors used stop codons to generate deletions instead of simply deleting the sequence of interest. However, the introduced stop codons can sometimes lead to a read-through effect, which may vary in its degree depending on the surrounding DNA sequence. Therefore, it is worth mentioning whether the authors have taken this possibility into account.

We thank the Reviewer for pointing this out, we agree that is an important point and needs to be mentioned. The following paragraph was added to the Methods: “The introduction of a stop codon was considered equivalent to a deletion mutant because in the cases in which the two approaches were compared, we obtained the same results. The finding that key single mutations in the FL protein mimic the effect of a stop-codon mediated deletion further suggested to us that the rare event of a read through over a stop codon does not occur in our conditions. This is also consistent with the observation that the introduction of the stop codon always generated a single distinct phenotype, ruling out the possibility of a leaky read through phenotype. Finally, functional data obtained with the stop codon-induced deletions in the HEK293 cell environment generated the same fold-change in cAMP affinity measured using truncated proteins in ITC”.

5. In line 207, “presumably because the channel is already saturated at the endogenous cAMP concentration of the cell line”. The reason that hHCN1 is more right-shifted than hHCN2 or hHCN4 is not well understood and could be due to other reasons, not because of the cAMP binding of hHCN1 is saturated. I suggest remove this explanation.

We agree with this observation, raised also by Reviewer 1, that the reasons why HCN1 $V_{1/2}$ is right-shifted are not fully understood. Here we exploit the known finding that HCN1 is pre-bound to cAMP when expressed in HEK293 cells (or in *E. coli*, Lolicato et al 2011) and that mutations like R549E (this study and Saponaro et al, eLife 2018, Porro et al, eLife 2019) or addition of peptides, like TRIP8bnano (Saponaro et al, eLife 2018,), induce a left shift by reducing CNBD affinity for cAMP. We have modified the sentence as follows:

“The same experiment was repeated with hHCN1 (Figure 5b). In this case, we exploited the known

finding that in HEK293 cells a reduction in cAMP affinity typically shifts the HCN1 $V_{1/2}$ value to the left. In this case, the mutation R549E, known to reduce cAMP affinity by 1000-fold¹³, induced a -8 mV shift (FL= -73.1 ± 0.4 , R549E= -81.4 ± 0.9) (Fig. 5b,c). Similarly, deletion of αE (ΔE), or neutralization of the salt bridge partner aspartate (D671A in mHCN2, D629A in hHCN1), caused a left shift of about 5 mV in $V_{1/2}$ (ΔE = -78.6 ± 0.8 , D629A= -77.5 ± 0.6).”

Minor points:

1. Line 269, “mV” was missing for ΔE .
2. Line 354, citation is in a different style.
- 3, Line 121, Clinker should be C-linker.
- 4, Line 208, typo of “typically”

Thank you for pointing out these errors, we have introduced the corrections.

Reviewers' Comments:

Reviewer #1:

Remarks to the Author:

Overall, I think the revised manuscript seems much clearer to me. However, there are some issues in the manuscript and response that are not clear to me.

I am still not clear on how the difference between the K_d value for cAMP binding to the isolated HCN2 CNBD and $K_{1/2}$ of cAMP effect on HCN2 has been resolved or how the affinity in the D'E' switch explains this? Parts of the manuscript where this difference is discussed or alluded to are not that clear to me. I think a more straight forward finding here is that the D' and E' helices appear to increase the potency of cAMP in the HCN4 isoform and to increase the affinity of cAMP for the isolated CNBD of HCN2. The data using one concentration of cAMP and measurement of a shift on HCN2 and HCN1 in HEK cells seems more tenuous without a complete concentration response curve for the wild type and mutants, as has been carried out for the HCN4 isoform. This is especially true for the HCN2 isoform where it seems that one micromolar cAMP is not saturating, but 15 micromolar may be required, in your HEK cells as compared to the K_d values as measured by ITC are 0.3 micromolar for the delta C-term construct. More detail and comments are below.

"It is unclear why the isolated cyclic nucleotide binding domain (CNBD) displays in vitro ten to hundred times less affinity for cAMP than the full-length channel in patch experiments (micromolar vs nanomolar). Here we resolve address the incongruity by showing that HCN channels are endowed with an affinity switch for cAMP, so far overlooked."

Has this been resolved? I note that you mention that you have determined a concentration versus shift experiment relationship using HCN2 in HEK cells, but this requires 15 micromolar to saturate. A complete concentration curve is not presented but it does not seem like there is a big difference between the K_d values obtained by ITC and the $K_{1/2}$ values obtained by patch clamp for the HCN2 isoform (or the HCN4 isoform as I mentioned previously).

The authors mention that a direct comparison between absolute values of $K_{1/2}$ and K_d obtained patch clamp and ITC, respectively, is not informative. Yet, isn't this what you have proposed to resolve in your statement above, why it is that the affinity of the isolated CNBD is ten to one hundred times less than the $K_{1/2}$ measured by patch clamp? This is a bit confusing.

"We work in whole-cell patch clamp from HEK293 cells and we have determined (but not published) the dose-response curve obtained by plotting the $V_{1/2}$ shift vs cAMP concentration (in the pipette) for mHCN2."

If you have these data, why are they not included and compared with your ITC data using the isolated HCN2 CNBD? This comparison seems to make the most sense? Alternatively, you could carry out ITC in the HCN4 isolated CNBD?

I also understand that you would like know if the affinity switch is found in the other HCN isoforms. However, with only one concentration of cAMP, I think it is difficult to extend your conclusions to $K_{1/2}$ for full-length HCN1 and HCN2. Again, I think the variability of the data around the mid-point of the concentration response curve (1 micromolar), without data from the upper and lower concentrations makes the conclusion much less certain for HCN2 and especially HCN1 where there is no ITC data.

"Why the values differ between cellular contexts or methods (whole cell vs inside out) and between HCN subtypes is a very good question but beyond the scope of this study."

I agree that the reason these differences exist is beyond the scope of your study. But these

differences make it more difficult to make conclusions about the relationship between binding affinity of cAMP to the isolated CNBD and the potency of cAMP in the full-length channel. Again, I think that the data presented clearly show that the D' and E' helix control cAMP potency in HCN4 and binding affinity of cAMP to the isolated CNBD.

I previously commented on the relationship between K_d and EC_{50} or $K_{1/2}$. The authors responded, in part, below.

"We agree on the above, and in our opinion a way to summarize the whole point is that by patch clamp we measure "efficacy" of cAMP, while in ITC, we measure binding, i.e. "affinity" of cAMP to the CNBD."

I think the authors mean potency ($K_{1/2}$, EC_{50}) and not efficacy? I think efficacy usually refers to the maximum effect?

Also, the authors replied "There is no way to directly compare the two values." I assume this refers to $K_{1/2}$ and K_d ? Again, if there is no way to compare them directly, then it would be difficult to resolve the question raised in the abstract, that the K_d of binding of cAMP to the isolated CNBD is ten to one hundred times the value of $K_{1/2}$ obtained by patch clamp?

As I mentioned previously, this may be due to the fact that the $K_{1/2}$ value reflects factors in addition to and beyond the binding affinity of the initial binding step? Some discussion of how K_d and $K_{1/2}$ are related might be helpful because their comparison is really the main part of this paper?

In regard to figure 8 and your response to it.

"Because the concentration of applied fluorescently tagged cAMP is kept constant during the voltage pulse, the increase in fluorescence intensity at the membrane patch must result from an increase in binding affinity."

My understanding of these kinds of experiments is that a change in fluorescence indicates a change in the environment surrounding the CNBD. Why is this change in environment interpreted as an increase in binding affinity instead of a decrease in binding affinity? I am not clear on this and it might help to clarify this in this section? Could this change in fluorescence be due to the initial movement of the voltage-sensor rather than the opening of the pore?

The following phrase, "To address both these questions at the same time," which addresses why the HCN2 isoform was used for the ITC. I am not sure this is the best rationale for using HCN2 rather than HCN4?

Line 160 "Since the differences in affinity of the hHCN2 CNBDs mirrored the results obtained by electrophysiology with rbHCN4 channels, as well as with mHCN2 where the $K_{1/2}$ measured in oocyte inside out patches is about 8-fold higher for the construct corresponding to $\Delta DE'$ compared to wildtype values we can conclude that αDE helices affect the affinity rather than the efficacy of cAMP binding and that the effect is conserved across HCN subtypes. "

I think this is comparing $K_{1/2}$ data from the Siegelbaum group (full-length HCN2) and the Zagotta group (where a shorter HCN2 construct was used)? This could provide good evidence in favor of your ideas and data. I missed this the first time I read your manuscript. I think this comparison, using a similar system (patches excised from *Xenopus* oocytes), albeit in different labs, to measure the effect of cAMP is good support for your findings. It might help to spend a bit more time to detail this difference more clearly and perhaps earlier in your manuscript. For example, what were the values for $K_{1/2}$ for these two studies? It still does not resolve completely why the effect of cAMP on HCN2 is less potent in HEK cells.

Also in line 160, what is meant by efficacy? Is it the maximum effect in this case?

"We show in Figure 3, that the content of cAMP in the protein matches the changes in N. See also Lolicato et al, 2011, where we use the same approach."

Figure 3 shows that there is cAMP bound to the CNBD. But where does it show that it matches the change in N? I understand "matches" to mean it would quantitatively explain the reduced stoichiometry. In order to do this, you might have to re-calculate the amount of cAMP versus protein, plot versus the heat measured by ITC and re-fit? If this is done, is the value for stoichiometry now one? Without a quantitative relationship, it might be clearer to say that the already-bound cAMP and misfolded protein may be responsible for the low value of stoichiometry.

Line 144. "While these results clearly highlight a crucial role of α DE in regulating the response of rbHCN4 to cAMP, two main aspects need to be investigated further: i) Whether the helices affect the affinity of the CNBD for cAMP, or the efficacy of the ligand in facilitating channel gating; ii) Whether α DE have a similar effect in other HCN subtypes."

Again, I am not sure what efficacy refers to here? Do you mean potency ($K_{1/2}$) and efficacy means maximum effect?

Again, in the absence of direct comparisons of K_d measured by ITC and $K_{1/2}$ measured by patch clamp in the same isoform, I think that the idea that the α D'E" enhances potency by enhancing affinity is not as well supported.

Line 369 "Our ITC data demonstrate that the addition of DE helices to the HCN2 CNBD fragment moves the affinity to the nanomolar range, mirroring patch clamp results."

$K_{1/2}$ for either HCN2 or HCN4 is not in the nanomolar range in your study?

Line 409 "the locking system of α DE is not required for the action of the voltage sensor to modulate cAMP affinity works independently from the voltage sensor."

I think it might be better to say that the change in fluorescence of the cAMP analog upon hyperpolarization is not greatly modified by the D' and E' helices which suggests that their interactions with the CNBD are independent?

There are several typos in the manuscript.

e.g. "alfa helices"

Reviewer #2:

Remarks to the Author:

The authors have satisfactorily addressed all of my concerns.

REPLY TO THE REVIEWER COMMENTS

Reviewer #1 (Remarks to the Author):

Overall, I think the revised manuscript seems much clearer to me. However, there are some issues in the manuscript and response that are not clear to me.

I am still not clear on how the difference between the K_d value for cAMP binding to the isolated HCN2 CNBD and $K_{1/2}$ of cAMP effect on HCN2 has been resolved or how the affinity in the D'E' switch explains this? Parts of the manuscript where this difference is discussed or alluded to are not that clear to me. I think a more straight forward finding here is that the D' and E' helices appear to increase the potency of cAMP in the HCN4 isoform and to increase the affinity of cAMP for the isolated CNBD of HCN2. The data using one concentration of cAMP and measurement of a shift on HCN2 and HCN1 in HEK cells seems more tenuous without a complete concentration response curve for the wild type and mutants, as has been carried out for the HCN4 isoform. This is especially true for the HCN2 isoform where it seems that one micromolar cAMP is not saturating, but 15 micromolar may be required, in your HEK cells as compared to the K_d values as measured by ITC are 0.3 micromolar for the delta C-term construct. More details and comments are below.

1-If we compare the results of ITC on HCN2 (Saponaro et al, PNAS 2014 and this manuscript) with the data in the literature on measurements of HCN2 in inside-out in oocytes (Chen et al, 2001; Ulens and Siegelbaum, 2003; Zagotta et al 2003), there is full coherence with the role of the D and E helices in determining the different affinities. Indeed, the FL HCN2 has nanomolar affinity and upon removal of the C terminus (including the first four aa of the D helices as in the construct $\Delta DE'$ presented in this work) the channel becomes from 4 to 6 less sensitive to cAMP (Zagotta 2003). This is in line with the finding shown here where the affinity changes 4-fold in ITC.

But, when we take as an example HCN4 or HCN2 measured in whole cell in HEK cells, then the effect of the helices remains valid but only in relative (fold changes) rather than in absolute terms. Indeed, for some yet unknown reasons, the $K_{1/2}$ of the channels measured in these conditions are (surprisingly) in the micromolar range rather than in nanomolar range. In the case of HCN4, even the recording in inside-out (in HEK cells) shows micromolar affinity (see Baruscotti et al, Eur Heart J 2017, now quoted in the ms). This finding is remarkably different from the nanomolar value (200 nM) originally described for the native I_f current in sinoatrial node myocytes (DiFrancesco and Tortora, Nature, 1991). Despite these discrepancies in the absolute affinity values, we show here that removal of the D and E helices decreases HCN4 and HCN2 response to cAMP by 30-fold, as expected (see the dose-response curve of HCN2, now included in the paper as Figure 3B).

In this case, therefore, we do agree with the Reviewer that our data do not fully resolve this issue of absolute concentrations, but our data show that in all cases, helices D and E are responsible for a 30-fold switch in affinity and efficacy.

We have consequently reworded the Abstract and the Introduction, clarifying the general message, as above. The main message of the manuscript is that folding/unfolding of the D and E helices is a master switch which determines either a low or high affinity of the CNBD for cAMP binding. A 30-fold increase in affinity measured in ITC is also paralleled by an identical 30-fold increase in efficacy measured in the full-length channel by patch clamp.

“It is unclear why the isolated cyclic nucleotide binding domain (CNBD) displays in vitro ten to hundred times less affinity for cAMP than the full-length channel in patch experiments (micromolar vs nanomolar). Here we resolve address the incongruity by showing that HCN channels are endowed with an affinity switch for cAMP, so far overlooked.”

Has this been resolved? I note that you mention that you have determined a concentration versus shift experiment relationship using HCN2 in HEK cells, but this requires 15 micromolar to saturate. A complete concentration curve is not presented but it does not seem like there is a big difference between the K_d values obtained by ITC and the $K_{1/2}$ values obtained by patch clamp for the HCN2 isoform (or the HCN4 isoform as I mentioned previously).

2-The complete dose response curve of HCN2 is now included in Figure 5b. Data show that DE deletion reduces the $K_{1/2}$ by about 30-fold (27-fold to be precise).

The authors mention that a direct comparison between absolute values of $K_{1/2}$ and K_d obtained patch clamp and ITC, respectively, is not informative. Yet isn't this what you have proposed to resolve in your statement above, why it is that the affinity of the isolated CNBD is ten to one hundred times less than the $K_{1/2}$ measured by patch clamp? This is a bit confusing.

See above, reply to point n.1

“We work in whole-cell patch clamp from HEK293 cells and we have determined (but not published) the dose-response curve obtained by plotting the $V_{1/2}$ shift vs cAMP concentration (in the pipette) for mHCN2.”

If you have these data, why are they not included and compared with your ITC data using the isolated HCN2 CNBD? This comparison seems to make the most sense? Alternatively, you could carry out ITC in the HCN4 isolated CNBD?

See above, reply to point n.2

I also understand that you would like know if the affinity switch is found in the other HCN isoforms. However, with only one concentration of cAMP, I think it is difficult to extend your conclusions to $K_{1/2}$ for full-length HCN1 and HCN2. Again, I think the variability of the data around the mid-point of the concentration response curve (1 micromolar), without data from the upper and lower concentrations makes the conclusion much less certain for HCN2 and especially HCN1 where there is no ITC data.

See above, reply to point n.2

“Why the values differ between cellular contexts or methods (whole cell vs inside out) and between HCN subtypes is a very good question but beyond the scope of this study.”

I agree that the reason these differences exist is beyond the scope of your study. But these differences make it more difficult to make conclusions about the relationship between binding affinity of cAMP to the isolated

CNBD and the potency of cAMP in the full-length channel. Again, I think that the data presented clearly show that the D' and E' helix control cAMP potency in HCN4 and binding affinity of cAMP to the isolated CNBD.

3. Our data also show that the presence/absence of D and E helices induces a 30-fold change in affinity and in efficacy, in different subtypes and independent of the cell context or configuration used.

I previously commented on the relationship between K_d and EC_{50} or $K_{1/2}$. The authors responded, in part, below.

“We agree on the above, and in our opinion a way to summarize the whole point is that by patch clamp we measure “efficacy” of cAMP, while in ITC, we measure binding, i.e. “affinity” of cAMP to the CNBD.”

I think the authors mean potency ($K_{1/2}$, EC_{50}) and not efficacy? I think efficacy usually refers to the maximum effect?

4. We understand that the terminology suggested by the reviewer is the one frequently used in pharmacology. However, we prefer to use the terminology from a seminal paper in the field of ion channels biophysics by Colquhoun (Br J Pharmacology 1998) that defines affinity and efficacy as follow: “...*affinity (...)* is simply the microscopic equilibrium (or rate) constant(s) for binding to the inactive state(s). *Efficacy is everything else. So efficacy is simply the set of all of the other microscopic equilibrium (or rate) constants, which describe all the transduction events that follow the initial binding reaction.*”

We find this description most suitable for our study and, with the appropriate reference to the Colquhoun paper in the introduction, we would prefer to leave as it is.

Also, the authors replied, “There is no way to directly compare the two values.” I assume this refers to $K_{1/2}$ and K_d ? Again, if there is no way to compare them directly, then it would be difficult to resolve the question raised in the abstract, that the K_d of binding of cAMP to the isolated CNBD is ten to one hundred times the value of $K_{1/2}$ obtained by patch clamp?

5. The sentence in the abstract has been removed.

As I mentioned previously, this may be due to the fact that the $K_{1/2}$ value reflects factors in addition to and beyond the binding affinity of the initial binding step? Some discussion of how K_d and $K_{1/2}$ are related might be helpful because their comparison is really the main part of this paper?

6. We fully agree, and we believe that the definition of Colquhoun of “efficacy” perfectly matches the concept expressed by the Reviewer. This was why we had originally chosen to use his terminology.

In regard to figure 8 and your response to it.

“Because the concentration of applied fluorescently tagged cAMP is kept constant during the voltage pulse, the increase in fluorescence intensity at the membrane patch must result from an increase in binding affinity.”

My understanding of these kinds of experiments is that a change in fluorescence indicates a change in the

environment surrounding the CNBD. Why is this change in environment interpreted as an increase in binding affinity instead of a decrease in binding affinity? I am not clear on this and it might help to clarify this in this section? Could this change in fluorescence be due to the initial movement of the voltage-sensor rather than the opening of the pore?

7. With respect, the Reviewer is wrong in the interpretation of these experiments. The fluorescence signal is not related to an environmental change around the CNBD but to a gain in the fluorescent cAMP analog. The latter occurs when the cAMP analog moves from the aqueous environment into the CBND binding pocket. The rationale of the patch clamp fluorimetry in HCN channels is described in detail in a series of publications [that we have quoted for the interested reader]. We find it not necessary to re-explain the method in the context of this paper.

The following phrase, “To address both these questions at the same time,” which addresses why the HCN2 isoform was used for the ITC. I am not sure this is the best rationale for using HCN2 rather than HCN4?

8. We agree that this might be confusing. The sentence has been removed.

Line 160 “Since the differences in affinity of the hHCN2 CNBDs mirrored the results obtained by electrophysiology with rbHCN4 channels, as well as with mHCN2 where the $K_{1/2}$ measured in oocyte inside out patches is about 8-fold higher for the construct corresponding to $\Delta DE'$ compared to wildtype values we can conclude that αDE helices affect the affinity rather than the efficacy of cAMP binding and that the effect is conserved across HCN subtypes. “

I think this is comparing $K_{1/2}$ data from the Siegelbaum group (full-length HCN2) and the Zagotta group (where a shorter HCN2 construct was used)? This could provide good evidence in favor of your ideas and data. I missed this the first time I read your manuscript. I think this comparison, using a similar system (patches excised from *Xenopus* oocytes), albeit in different labs, to measure the effect of cAMP is good support for your findings. It might help to spend a bit more time to detail this difference more clearly and perhaps earlier in your manuscript. For example, what were the values for $K_{1/2}$ for these two studies? It still does not resolve completely why the effect of cAMP on HCN2 is less potent in HEK cells.

9. We have followed the reviewer suggestion and have highlighted this point better in the introduction.

Also in line 160, what is meant by efficacy? Is it the maximum effect in this case?

See reply to question n. 4.

“We show in Figure 3, that the content of cAMP in the protein matches the changes in N. See also Lolicato et al, 2011, where we use the same approach.”

Figure 3 shows that there is cAMP bound to the CNBD. But where does it show that it matches the change in N? I understand “matches” to mean it would quantitatively explain the reduced stoichiometry. In order to do this, you might have to re-calculate the amount of cAMP versus protein, plot versus the heat measured by ITC and re-fit? If this is done, is the value for stoichiometry now one? Without a quantitative relationship, it might be clearer to say that the already-bound cAMP and misfolded protein may be responsible for the low value of stoichiometry.

10. We are sorry to say that we don't find where we have stated that "cAMP bound matches the change in N". We do show in Figure 3 and comment in the text that the high content of cAMP inside the fragments with helices D and E, indicates a higher affinity induced in the CNBD by the helices.

Line 144. "While these results clearly highlight a crucial role of α DE in regulating the response of rbHCN4 to cAMP, two main aspects need to be investigated further: i) Whether the helices affect the affinity of the CNBD for cAMP, or the efficacy of the ligand in facilitating channel gating; ii) Whether α DE have a similar effect in other HCN subtypes."

Again, I am not sure what efficacy refers to here? Do you mean potency ($K_{1/2}$) and efficacy means maximum effect?

See reply to question n. 4.

Again, in the absence of direct comparisons of K_d measured by ITC and $K_{1/2}$ measured by patch clamp in the same isoform, I think that the idea that the alpha D'E" enhances potency by enhancing affinity is not as well supported.

11. We interpret the same fold shift in affinity and efficacy obtained in the presence/absence of the helices, as **strong causal relationship between the two parameters**. We have added this reasonable assumption in the text (discussion).

"While we cannot explain in absolute terms the changes in affinity and efficacy related to the helices, the identity of the fold changes induced by their removal/addition leaves no doubt that they are causally coupled."

Nonetheless, as the Reviewer also agreed upon, there is experimental evidence in the case of HCN2, in which even the absolute numbers underscore this finding (comparison of Siegelbaum group and Zagotta group data for $K_{1/2}$ and comparison in this work of ITC data for K_D).

Line 369 "Our ITC data demonstrate that the addition of DE helices to the HCN2 CNBD fragment moves the affinity to the nanomolar range, mirroring patch clamp results."

$K_{1/2}$ for either HCN2 or HCN4 is not in the nanomolar range in your study?

12. Yes, the reviewer is correct in that our data from patch clamp recordings in whole cell in HEK cells do not show nanomolar values. But such nanomolar values have been reported in the literature when the same kind of experiments were performed in oocytes in inside-out configuration. We refer to these data with the respective reference.

Line 409 "the locking system of α DE is not required for the action of the voltage sensor to modulate cAMP affinity works independently from the voltage sensor."

I think it might be better to say that the change in fluorescence of the cAMP analog upon hyperpolarization is not greatly modified by the D' and E' helices which suggests that their interactions with the CNBD are independent?

13. The interpretation suggested by the Reviewer is not coherent with the method, so we cannot use it.

There are several typos in the manuscript.

e.g. "alfa helices"

14. Thank you for pointing it out, we have corrected it.

Reviewer #2 (Remarks to the Author):

The authors have satisfactorily addressed all of my concerns.

Reviewers' Comments:

Reviewer #1:

Remarks to the Author:

Thank you for your very clear responses to my latest set of questions and comments.

I noted a couple of minor typos during my reading of the manuscript.

Figure 5 "helix D and E controls cAMP affinity in HCN2" - 'control' without the 's'?

"relevant in this context is the experimental finding that the isolated CNBD can switch its affinity for cAMP by order of magnitude." – an order of magnitude?